# “Smart” Antimicrobial Nanocomplexes with Potential to Decrease Surgical Site Infections (SSI)

**DOI:** 10.3390/pharmaceutics12040361

**Published:** 2020-04-15

**Authors:** Zehra Edis, Samir Haj Bloukh, May Reda Ibrahim, Hamed Abu Sara

**Affiliations:** 1Department of Pharmaceutical Sciences, College of Pharmacy and Health Sciences, Ajman University, Ajman PO Box 346, UAE; may.ibrahim@ajman.ac.ae; 2Department of Clinical Sciences, College of Pharmacy and Health Sciences, Ajman University, Ajman PO Box 346, UAE; s.bloukh@ajman.ac.ae (S.H.B.); h.abusara@ajman.ac.ae (H.A.S.)

**Keywords:** antimicrobial activity, surgical suture, biodegradable, surgical site infection, silver nanoparticles, trans-cinnamic acid, natural cinnamon extract, povidone iodine, polyiodides

## Abstract

The emergence of resistant pathogens is a burden on mankind and threatens the existence of our species. Natural and plant-derived antimicrobial agents need to be developed in the race against antibiotic resistance. Nanotechnology is a promising approach with a variety of products. Biosynthesized silver nanoparticles (AgNP) have good antimicrobial activity. We prepared AgNPs with *trans*-cinnamic acid (TCA) and povidone–iodine (PI) with increased antimicrobial activity. We synthesized also AgNPs with natural cinnamon bark extract (Cinn) in combination with PI and coated biodegradable Polyglycolic Acid (PGA) sutures with the new materials separately. These compounds (TCA-AgNP, TCA-AgNP-PI, Cinn-AgNP, and Cinn-AgNP-PI) and their dip-coated PGA sutures were tested against 10 reference strains of microorganisms and five antibiotics by zone inhibition with disc- and agar-well-diffusion methods. The new compounds TCA-AgNP-PI and Cinn-AgNP-PI are broad spectrum microbicidal agents and therefore potential coating materials for sutures to prevent Surgical Site Infections (SSI). TCA-AgNP-PI inhibits the studied pathogens stronger than Cinn-AgNP-PI in-vitro and on coated sutures. Dynamic light scattering (DLS), ultraviolet-visible spectroscopy (UV-Vis), Fourier Transform infrared spectroscopy (FT-IR), Raman, X-ray diffraction (XRD), microstructural analysis by scanning electron microscopy (SEM) and energy dispersive spectroscopy (EDS) confirmed the composition of TCA-AgNP-PI and Cinn-AgNP-PI. Smart solutions involving hybrid materials based on synergistic antimicrobial action have promising future perspectives to combat resistant microorganisms.

## 1. Introduction

Antimicrobial resistance is, according to the World Health Organization (WHO), one of the major threats to the existence of mankind [1]. Increase in uncontrolled, irrational antibiotic use, lack of incentives, and research efforts for the development of new antibiotics resulted in antibiotic resistance worldwide [2]. ESKAPEE pathogens (*Enterococcus faecium*, *Staphylococcus aureus*, *Klebsiella pneumoniae*, *Acinetobacter baumannii*, *Pseudomonas aeruginosa*, *Enterobacter spp*., *and Escherichia coli*) play a prominent role in this global epidemic and are considered by the WHO as high-risk pathogens [3,4,5,6]. Alone in the EU antibiotic resistant, microbes cause reportedly disability, mortality, and increasing costs in medical care [7,8,9]. In the recent history, there is still no single multi-drug-resistant microorganism capable of causing a pandemic like the coronavirus disease 2019 (COVID-19) [10]. Nevertheless, this global outbreak is just a glimpse into similar scenarios in the near future, if we fail to develop solutions against antimicrobial resistant pathogens. Methicillin-resistant *Staphylococcus aureus* (MRSA) lead to 20,000 related deaths in the United States in 2017 only and remains to be one of the important infection-associated mortality reasons [11,12]. The formation of biofilms on biotic and abiotic surfaces [13] paved the way for biofilm antibiotic tolerance as another major health threat [14,15]. This additional problem hampers the treatment of infections, most often gives rise to chronic infections, in the worst case to therapeutic failure, severe morbidity, and mortality [4]. More than 80% of microbial infections are caused by microbial biofilm formation and pose a severe health risk because treatment with antibiotics remains ineffective [16]. The incidence of chronic wound infections is increasing globally to alarming levels [17]. Chronic wounds are marked by high bacterial colonization and infection rates, which can be treated only by novel therapeutic approaches other than the use of conventional antibiotics [18,19]. Chronic wound tissues can contain microbiomes of complex communities with coexisting fungal and bacterial colonies of different strains within biofilms [20]. According to this inter-kingdom model, fungi like *Candida albicans* can offer structures for other bacteria (*S. aureus*, *P. aeruginosa*, *Streptococcus* spp., *E. faecalis*, *E. coli*, and *Acinetobacter baumannii* etc.) to attach, form intricate biofilms, and increase the resistance against any form of treatment [20]. Exploring plant-, polymer-, and nanotechnology-based therapeutic approaches rendered a plethora of promising results. Within the year 2019 alone, publications of medicinal plant-related research articles skyrocketed due to the huge interest of the scientific community to explore suitable therapeutic agents [21]. Targeting bacterial virulence factors like biofilm formation, quorum sensing, or motility instead of the viability is another interesting alternative in the current post-antibiotic-era [22]. These efforts can render successful applications in wound healing, infection control and surgical wound closure. Currently, infections on wounds lead in the worst-case scenario to Surgical Site Infections (SSI) with increased treatment costs, higher hospitalization rate, longer treatment duration, severe morbidity, and mortality [23,24]. 

Polymer, biodegradable surgical sutures are widely used for wound closure [24,25]. Polyglycolic acid (PGA) is a synthetic, multifilamented, braided suture and is absorbed within 60 to 90 days by hydrolysis from the surrounding tissues [24]. This linear, aliphatic polyester is a polymer of glycolid acid, non-toxic, non-irritating, and low-cost material [24]. These properties of PGA are needed for a successful wound healing process but not enough to stop SSI due to bacterial resistance against antibiotics and bactericides [24]. Surgical sutures are a foreign material in the body, and as such cause tissue responses leading to inflammation and further complications [25]. PGA is a multifilament suture, and due to this capillarity offers more surface for microbial attachment compared to monofilament sutures [25]. This affinity to microbial growth results in biofilm formation, higher tissue response, and increased infection rate [25]. An alternative approach is to coat the surgical sutures with antimicrobial agents and deliver the drug directly on-site, which leads to faster healing without complications [25]. Additionally, 23% of SSI are associated with *S. aureus* biofilm formation on surfaces of sutures and implants [25]. Coating PGA with antibiotics, therapeutic agents, nanoparticles, and biocidal compounds increases its biological activity, reduces biofilm formation and contamination [25]. Dip-coated PGA can deliver the antimicrobial compounds by controlled drug-release directly on the target tissue [25].

Iodine has been used as a biocidal agent in many different applications for a long time [26,27,28,29]. Iodine is used already in wound care but has few disadvantages due to sublimation and skin irritation [26,27,30,31,32]. Nevertheless, the addition of iodine may reduce SSI by improving antimicrobial properties of dip-coated PGA. Sublimation is directly related to uncontrolled iodine release, increased skin irritation, and short-term antimicrobial activity of the drug carrier. This problem can be solved by using complex cation stabilized polyiodides instead of iodine within the wound dressing or surgical sutures [28]. Polyiodides consist of [I_2k+n_]^n−^ units and are stabilized by donor–acceptor interactions of the combined iodine molecules and iodide ions [33,34,35]. Triiodides are the smallest examples of polyiodides and contain either symmetric or asymmetric [I-I-I]^−^ groups, which are connected to each other by halogen and/or hydrogen bonding [36,37] Triiodides were successfully used in combination with phototherapy in wound healing [38,39]. Ideal, symmetric triiodide units are characterized by pure halogen bonding and can be identified by characteristic Raman stretching vibrations at 108 cm^−1^ due to the electron density redistribution during the formation of halogen bonds [27,40,41]. The covalent character of I_2_ units is reduced due to their strong involvement in the halogen bonding [40]. Symmetric triiodide units with halogen bonding character are more stable than asymmetric triiodide units and can be used as antimicrobial agents [42,43]. The stability and antimicrobial properties of such symmetric triiodide units remains high, even if incorporated into other polyiodide moieties like pentaiodides [40,44]. 

Polymers can stabilize iodine, decrease the sublimation rate, increase long-term stability, durability, and controlled long-term release of iodine [45,46,47]. Povidone–iodine (PI) is a commercial disinfectant consisting of the polymer polyvinylpyrrolidone (PVP) and iodine [48]. PI is known for its broad-spectrum microbicidal properties and is used in wound care [32,47,48]. A recent paper from Schmitz et al. investigated PI in a randomized clinical trial with 101 enrolled patients [49]. It was found that 59.6% of the patients in the povidone iodine group reported adverse effects compared to 26.5% of patients in the standard care group without PI [49]. The adverse effects included burning/pain, pruritis, skin irritation around the wound and skin coloration [49]. These limited results indicate the need to improve PI delivery to the wound site. Goodwin et al. investigated the structure of povidone iodine and reported triiodide ions within the polymeric chain [50]. These triiodide ions are the reason for the release of free iodine and its antimicrobial properties. Reducing adverse effects can be accomplished by improved controlled-release mechanisms of iodine. A promising development is the combination of PVP with iodine nanoparticles and its incorporation into wound dressing materials to inhibit bacteria and biofilm formation which may reduce the above adverse effects due to minimizing the iodine content [31,48]. 

Nanotechnology offers a wide range of medical applications including biodegradable, plant and polymer-based antimicrobial compounds [51,52,53,54,55,56]. Nanomedicine includes organic and inorganic drug delivery systems, ensuring long shelf life with reduced drug loss and low toxicity [15,57]. Metallic nanoparticles are suitable pharmaceutical drug carriers with their large surface area to adsorb, bind, or entrap guest molecules like drugs, proteins, etc., and deliver them to the target area [57]. Especially silver nanoparticles (AgNP) are used in medical care products, cosmetics, food packaging, textiles, and cosmetics because of their antimicrobial activity [58,59]. Natural plants and their extracts can act as reducing and capping agents for silver ions. Facile and eco-friendly synthesis by green methods using natural plant products or their derivatives render biodegradable, biocompatible drug delivery solutions without the use of toxic chemicals [57]. Incorporating AgNPs into many products may have long-term effects on the environment, living-systems, and human health due to cytotoxic properties [60]. There is a need for improved, durable hybrid products of AgNPs and bio-enhancing plant extracts with controlled silver-/silver-ion release mechanisms [61]. Such bio-enhancers with decreased AgNP concentrations should optimize the antimicrobial activity, prevent the emergence of resistant mutants and minimize toxicity. Polymer nanocomposites can be used as wound dressings and release silver and silver ions by sustained release to the wound [60]. Incorporating iodine and PVP-I_2_ (povidone iodine, PI) as biocidal components can widen the antimicrobial potency and decrease the detrimental effects on environment and human health by minimizing AgNP content. 

Phytochemicals are another promising class of antimicrobial compounds and are successfully utilized as reducing and stabilizing agents for the green synthesis of AgNP [3,6]. *Cinnamomum zeylanicum* incorporates a plethora of different active natural ingredients including cinnamaldehyde, cinnamic acid and its derivatives with known potential of microbial growth control [62,63,64,65,66,67,68,69]. 

*Trans*-cinnamic acid is a hydrophobic, phenolic compound with antimicrobial, antioxidant, anti-cancer, and anti-inflammation properties [70]. Cinnamic acid in combination with gold nanoparticles achieves good results against different pathogens and brain-eating amoeba [71,72]. Due to its hydrobicity, trans-cinnamic acid needs to be incorporated into a hydrophilic carrier [70]. Letsididi et al. used nanoemulsion systems as carrier and proved antimicrobial-, antibiofilm activity of *trans*-cinnamic acid. Malheiro et al. investigated the antibacterial activity of cinnamaldehyde, cinnamic acid and cinnamic acid derivatives against *S. aureus* NCTC 10788 and *E. coli* NCTC 10418 by microdilution method [67]. Cinnamaldehyde showed the highest inhibition compared to cinnamic acid and its derivatives against the two pathogens. *E. coli* was more susceptible than *S. aureus* towards cinnamaldehyde [67]. Cinnamic acid had weak inhibitory action towards the microorganisms. Letsididi et al. compared the susceptibility of *S. aureus* and *P. aeruginosa CMCC 10104* (both strains from the China General Microbiological Culture Collection Center) against TCA nanoemulsion and pure TCA by agar well diffusion and dilution series [70]. The nano-emulsion of TCA was more effective than the pure TCA. *S. aureus* was strongly inhibited by the nanoemulsion of TCA with a concentration of 0.78 mg/mL. *P. aeruginosa* showed higher resistance to the nanoemulsion of TCA and was inhibited at a concentration of 3.13 mg/mL [70]. The nanoemulsion of TCA had more antibacterial activity against the Gram-positive *S. aureus* than the Gram-negative *P. aeruginosa*. Premkumar et al. synthesized AgNP with cinnamon extract and investigated its antimicrobial effects by disc diffusion methods against *S. aureus*, *P. aeruginosa* and *E. coli* (strains from the University of Madras, PGIBMS) [64]. *P. aeruginosa* and *S. aureus* were more susceptible against AgNP-cinnamon than *E.coli*. They compared antibacterial activity of AgNP reduced by cinnamon extract with AgNP alone and concluded higher susceptibility towards the biosynthesized AgNP [64]. 

In this work, we investigated combinations of hydrophobic *trans*-cinnamic acid (TCA), natural *cinnamomum zeylanicum* bark extract (Cinn) and hydrophilic povidone iodine (PI) to increase the antimicrobial effect of AgNPs. Our hypothesis is to optimize antimicrobial activity of AgNPs by combining silver nanoparticles with the known microbicidal agents TCA, Cinn, and PI. Reducing the AgNP content will decrease its hazards to human health and environment. The adverse effects related to PI and iodine can be minimized by this potent combination through controlled release of iodine [42,49,50]. The synergism of these compounds reduces the amount of needed PI for optimal microbial growth control and its adverse effects. We used *trans*-cinnamic acid for the chemical reduction and natural cinnamon bark extract for the biological reduction (green synthesis) of silver nitrate to AgNPs. TCA and *cinnamomum zeylanicum* bark extracts (Cinn) act also as reducing, capping, and stabilizing agents [64,67,68]. PI is a stabilizing agent and reservoir for triiodide anions releasing free molecular iodine for biocidal action [32,48]. AgNP is a drug carrier and adsorbs TCA, Cinn, and PI on its large surface [57,60]. TCA, Cinn, and PI as stabilizing agents prevent agglomeration of silver, resulting in smaller AgNPs and better microbial growth control. This hybrid antimicrobial agent/biocide/drug delivery system can be an advantage to prevent SSI and biofilm formation. The combinations of TCA, Cinn, AgNP, and PI were expected to have synergistic antimicrobial effects. Pure TCA and the Cinn bark extract were compared to investigate the effect of pharmacokinetic synergists in the natural plant extract Cinn with its manifold bioactive compounds [61,73]. Biogenic synthesis with plant extracts usually induce broad particle size distribution in most cases but can be also shape- and size determining [74,75]. We compared the particle size distribution of Cinn and TCA samples. The microstructural analysis confirmed the purity, particle size range and surface morphology of the products TCA-AgNP, TCA-AgNP-PI, Cinn-AgNP, and Cinn-AgNP-PI. 

The antimicrobial activities of our four samples TCA-AgNP, TCA-AgNP-PI, Cinn-AgNP, and Cinn-AgNP-PI and their dip-coated PGA sutures were tested against a total of 10 different microorganisms compared to four common antibiotics (chloramphenicol, cefotaxime, gentamycin, and nystatin). Gram-positive cocci included *Streptococcus pneumoniae (S. pneumoniae ATCC 49619)*, *Staphylococcus aureus (S. aureus ATCC 25923)*, and *Enterococcus faecalis* (*E. faecalis ATCC 29212).* Gram-negative bacteria were *Pseudomonas aeruginosa (P. aeruginosa WDCM 00026)* and *Escherichia coli (E. coli WDCM 00013).* The tested fungi were *Candida albicans* and *C. albicans WDCM 00054.* The dip-coated sutures were additionnaly tested against the Gram-positive *Streptococcus pyogenes (S. pyogenes ATCC 19615),* the spore-forming bacteria *Bacillus subtilis,* the Gram-negative bacteria *Proteus mirabilis* (*P. mirabilis ATCC 29906)*, and *Klebsiella pneumoniae (K. pneumoniae WDCM 00097)*. Our compounds exhibited microbicidal action. The Gram-negative pathogens *P. aeruginosa*, *E. coli*, *K. pneumoniae*, and the Gram-positve *S. aureus* were inhibited by TCA-AgNP-PI and its dip-coated PGA suture stronger than Cinn-AgNP-PI. 

## 2. Materials and Methods 

### 2.1. Materials

Silver nitrate, iodine (≥99.0%), potassium iodide, polyvinylpyrrolidone (PVP-K-30), and Mueller Hinton Broth (MHB) were obtained from Sigma Aldrich (Gillingham, UK). Disposable sterilized Petri dishes with Mueller Hinton II agar and McFarland standard sets were received from Liofilchem Diagnostici (Roseto degli Abruzzi (TE), Italy). Trans-cinnamic acid (TCA), povidone–iodine (PI) were purchased from Sigma-Aldrich Chemical Co. (St. Louis, MO, USA). Cinnamon sticks (*cinnamomum zeylanicum*) were obtained from the local market. The bacterial strains *S. pneumoniae* ATCC 49619, *S. aureus* ATCC 25923, *E. faecalis* ATCC 29212, *S. pyogenes ATCC 19615, P. mirabilis ATCC 29906,* the spore-forming bacteria *Bacillus subtilis* were purchased from Liofilchem (Roseto degli Abruzzi (TE), Italy). *E. coli* WDCM 00013 Vitroids, *P. aeruginosa* WDCM 00026 Vitroids, *K. pneumoniae* WDCM00097 Vitroids and *C. albicans* WDCM 00054 Vitroids were bought from Sigma-Aldrich Chemical Co. (St. Louis, MO, USA). Gentamicin (9125, 30 µg/disc), chloramphenicol (9128, 10 µg/disc), cefotaxime (9017, 30 µg/disc), and nystatin (9078, 100 IU/disc) were received from Liofilchem (Roseto degli Abruzzi (TE), Italy). Methanol (analytical grade) was obtained from Fisher Scientific (Loughborough, UK). All reagents were of analytical grade and used as received. Ultrapure water was used. Sterile, braided and absorbable polyglycolic acid (PGA) surgical sutures (DAMACRYL, 75 cm, USP:3-0, Metric:2, 19mm, DC3K19) were purchased from General Medical Disposable (GMD), GMD Group A.S., Istanbul, Turkey). 

### 2.2. Preparation of Silver Nanoparticles with Trans-Cinnamic Acid (TCA) as Capping Agent (TCA-AgNP)

TCA-AgNPs are prepared by reducing silver nitrate with sodium borohydride in the presence of TCA. Briefly, the TCA stock solution is dissolved in 10% methanol with the resulting concentration of 1 mM and silver nitrate stock solution is prepared in water with concentration of 5 mM. Then, 1 mL of 5 mM TCA is mixed with 15 mL of a 1 mM silver nitrate solution and the reaction mixture is stirred for 10 min. To prepare fresh reducing agent solution, the sodium borohydride is dissolved in water to obtain 4 mM aqueous solution and kept in an ice bath. 100 µL of this solution is added under stirring to the silver nitrate reaction mixture. The colorless solution converted into dark yellow color after adding sodium borohydride solution. The color change of the solution is due to the reduction of silver ions and the formation of TCA-AgNPs. The synthesized AgNPs are centrifuged at 4000 rpm for 30 min at room temperature (3K 30; Sigma Laborzentrifugen GmbH, Osterode am Harz, Germany). The supernatant part is discarded, and the precipitate is dispersed in distilled water by sonication for five minutes.

### 2.3. Preparation of Silver Nanoparticles with Cinnamon Bark Extract as Capping Agent (Cinn-AgNPs)

#### 2.3.1. Preparation of Cinnamomum zeylanicum (Cinn) Bark Extract

The *cinnamomum zeylanicum* bark extract is prepared similar to the method of Premkumar et al. [64]. *C. zeylanicum* bark was bought from a native shop, crushed into small pieces, and powdered in a blender to become even in size. The final separated powder is used for further studies. For the preparation of extract, 10 g of powder is added to a 500 mL Erlenmeyer flask with 200 mL sterile, distilled water and then boiled for 10 min. The mixture is filtered through Whatman filter paper No1 and the filtrate is kept in the fridge to be used within one week.

#### 2.3.2. Phytosynthesis of Nano-Scale Ag Particles (Cinn-AgNP)

Nano-scale AgNP from Cinn bark extract are obtained by adding AgNO_3_ (1 mmol) to 10% of aqueous Cinn extract under stirring at room temperature. The appearance of brown and opaque color in the solution indicates the formation of silver nanoparticles. To investigate the reaction time, AgNO_3_ is added into the aqueous extract plant and placed on a shaker at ambient temperature for 1 h. Finally, the absorption spectra of the samples are analyzed at 200–1000 nm. The AgNPs are purified by centrifugation at 4000 rpm for 30 min at room temperature. The supernatant is removed, and the precipitate is redispersed in distilled water by sonication for 1 min.

### 2.4. Synthesis of Povidone Iodine (PI) Capped Silver Nanoparticles (TCA-AgNP-PI and Cinn-AgNP-PI)

In a typical experiment, 1 mM of both samples of AgNPs capped with the TCA, and AgNPs photosynthesized by cinnamon extract, are added to 1mM of PI in a 200 mL flask at room temperature. The mixture is stirred until the total reaction is completed and analyzed by UV–Vis spectrometer in the range of 190–1000 nm.

### 2.5. Characterizations of Synthesized AgNP Samples

The compounds were analysed by dynamic light scattering (DLS), UV-Vis, FT-IR, Raman, X-ray diffraction (XRD), and microstructural analysis by SEM/EDS. These methods confirmed the composition of TCA-AgNP-PI and Cinn-AgNP-PI.

#### 2.5.1. Morphological Examination and Particle-Size Measurement

The AgNP samples were examined by scanning electron microscopy (SEM) equipped with energy-dispersive X-ray spectroscopy (EDX), model VEGA3 from Tescan (Brno, Czech Republic). A drop of each sample dispersed in distilled water was dropped onto a carbon-coated copper grid and left to dry under ambient conditions. The dried samples were coated by a gold film with Quorum Technology Mini Sputter Coater. The elemental composition was examined by EDS analysis system. 

#### 2.5.2. Size and Zeta-Potential Analysis

Dynamic light scattering (DLS) analysis was used to calculate the average size, size distribution and the polydispersity index (PDI) with model SZ-100 purchased from Horiba (Palaiseau, France). Furthermore, zeta (ζ)-potential measurement was done at room temperature to examine the dispersion and stability of the nanocolloidal samples.

#### 2.5.3. UV-Vis Spectrophotometry

To confirm the formation of AgNPs, the absorbance spectrum of different samples was recorded using a UV-Vis spectrophotometer model 1800 from Shimadzu (Kyoto, Japan) using the wavelength range from 190 to 800 nm.

#### 2.5.4. Fourier-Transform Infrared Spectroscopy

The four samples of AgNPs were freeze-dried and then studied at 400–4000 cm^−1^ using a FTIR spectrometer (Shimadzu, Koyoto, Japan). FTIR analysis was used to identify various organic molecules adhering to the metal nanoparticles.

#### 2.5.5. X-Ray Diffraction (XRD) Analysis of AgNPs

The structure and composition of our four products were analyzed by XRD (BRUKER, D8 Advance, Karlsruhe, Germany). Coupled Two Theta/Theta were used with Cu radiation (wavelength = 1.54060 A) and the time per step was 0.5 s with step size of 0.03.

#### 2.5.6. Characterization by Surface-Enhanced Raman Spectroscopy (SERS)

The analysis was carried out by a RENISHAW (Gloucestershire, UK) setup coupled with an optical microscope at room temperature. The excitation of solid-state laser beam (785 nm) was focused onto the sample through the 50× objective of a confocal microscope (Spot diameter—2micron). The sample solution was analyzed in a standard 1 cm × 1 cm cuvette and located in the pathway of the laser beam. The scattered light was collected and identified using a CCD-based monochromator, covering a spectral range of 50–3400 cm^−1^. Output power used was 10% and the spectral resolution of −1 cm^−1^ with the integration time—300 s.

### 2.6. Bacterial Strains and Culturing

The reference strains *S. pneumoniae* ATCC 49619, *S. aureus* ATCC 25923, *E. faecalis* ATCC 29212, *S. pyogenes* ATCC 19615, *E. coli* WDCM 00013 Vitroids, *P. aeruginosa* WDCM 00026 Vitroids, *K. pneumoniae* WDCM00097 Vitroids, and *C. albicans* WDCM 00054 Vitroids were used for the antimicrobial testing. These strains were stored at −20 °C and inoculated in MHB by adding the selected fresh bacteria and fungi into 10 mL Mueller Hinton broth. These prepared suspensions were kept at 4 °C until use. 

### 2.7. Investigation/Determination of Antibacterial and Antifungal Properties of TCA-AgNP, TCA-AgNP-PI, Cinn-AgNP, and Cinn-AgNP-PI

The four compounds TCA-AgNP, TCA-AgNP-PI, Cinn-AgNP, and Cinn-AgNP-PI were tested on Gram-positive *S. pneumoniae* ATCC 49619, *S. aureus* ATCC 25923, *E. faecalis* ATCC 29212, Gram-negative bacteria *E. coli WDCM 00013*, and *P. aeruginosa WDCM 00026.* The antifungal activities were tested on *C. albicans WDCM 00054*. 

#### 2.7.1. Procedure for Zone of Inhibition Plate Studies

The antimicrobial activities of TCA-AgNP, TCA-AgNP-PI, Cinn-AgNP, and Cinn-AgNP-PI against the mentioned pathogens were tested by the zone of inhibition plate method [76]. The bacteria were suspended in 10 mL Mueller Hinton Broth (MHB) and incubated at 37 °C for 2–4 h. *C. albicans* was incubated at 30 °C on Sabouraud Dextrose broth. A microbial culture of 100 μL adjusted to 0.5 McFarland standard was applied evenly with sterile cotton swabs on disposable, sterilized Petri dishes containing Mueller Hinton II agar (MHA). The prepared agar plates were dried for 10 min and used for the agar well- and disc-diffusion methods.

#### 2.7.2. Agar Well Diffusion Method

With a sterile well borer, a 6 mm diameter circular piece of MHB agar was removed from an already inoculated agar plate and filled with 20 mg of TCA-AgNP. Three other agar plates were each loaded in the same way with 20 mg TCA-AgNP-PI, Cinn-AgNP, and Cinn-AgNP-PI. AgNP, AgNP-PVP, TCA-AgNP, Cinn-AgNP, and selected antibiotics served as positive controls. These plates with the mentioned bacteria and the fungus *C. albicans WDCM 00054* were incubated for 24 h at 37 °C and 30 °C, respectively. The diameter of the zone of inhibition was measured with a ruler to the nearest millimeter. The antimicrobial susceptibility was based on the diameters of clear inhibition zone around the well. The microorganism is resistant to the tested compounds if there is no inhibition zone. The antimicrobial tests were replicated three times and the presented results are the average of three independent experiments.

#### 2.7.3. Disc Diffusion Method

Sterile filter paper discs (Himedia, India) with a diameter of 6 mm were impregnated with 2 mL of known concentrations of TCA-AgNP-PI, Cinn-AgNP and Cinn-AgNP-PI in pure water (50 µg/mL, 25 µg/mL, and 12.5 µg/mL). 

Antimicrobial tests were performed by disk diffusion method according to Clinical and Laboratory Standards Institute (CLSI) recommendations against the common antibiotics chloramphenicol, cefotaxime, gentamycin, and nystatin in the form of antibiotic discs [77]. The agar plates for the fungus *C. albicans WDCM 00054* were incubated for 24 h at 30 °C. The diameter of the zone of inhibition was measured with a ruler to the nearest millimeter. The antimicrobial susceptibility was based on the diameters of clear inhibition zone around the well. The microorganism is resistant to the tested compounds if there is no inhibition zone. The antimicrobial tests were replicated three times and the presented results are the average of three independent experiments.

### 2.8. Preparation of Sutures Coated with AgNPs Samples, Dip-Coating, and Characterization 

The four samples AgNPs capped with TCA (TCA-AgNP), AgNPs capped with TCA and coated with povidone–iodine (TCA-AgNP-PI), AgNPs capped with cinnamon bark extract (Cinn-AgNP), and AgNPs capped with the cinnamon bark extract and surrounded by povidone iodine (Cinn-AgNP-PI) were used to impregnate sutures. Multifilamented, sterile, uncoated, and braided PGA sutures (DAMACRYL from GMD Group A.S. with diameter (19 mm), corresponding to the United States Pharmacopeia standard USP3-0), were dip-coated with the four compounds separately. The coating process of the sutures is as follows: 10 suture fragments of approximately 2.5 cm (treated with acetone and dried at room temperature), 10 mL of cell-free supernatant and 50 mL of TCA-AgNP, TCA-AgNP-PI, Cinn-AgNP, and Cinn-AgNP-PI solutions (1 mM) were incubated at 25 °C and stirred at 130 rpm for 18 h. Visual color change was observed and the color change (from blue to brownish yellow) of sutures indicated the presence of AgNPs. Synthesized AgNP-coated sutures were removed from the flasks and dried for 18 h under ambient temperature until the characterization and antimicrobial assays. These coated sutures were analyzed by SEM to reveal the morphological changes. In-vitro antimicrobial performance of these dip-coated sutures was tested by the zone of inhibition assay against *S. pneumoniae ATCC 49619*, *S. aureus ATCC 25923*, *E. faecalis ATCC 29212*, *S. pyogenes ATCC 19615*, the spore-forming bacteria *Bacillus subtilis*, the Gram-negative bacteria *E. coli WDCM 00013*, *P. aeruginosa WDCM 00026*, *P. mirabilis ATCC 29906*, and *K. pneumoniae WDCM 00097.* The antifungal activities were tested on *C. albicans WDCM 00054*. 

### 2.9. Statistical Analysis

Data are expressed as mean ± standard deviation. The statistical significance between groups is calculated by one-way ANOVA and a value of *p* < 0.05 is considered statistically significant. The statistical analysis was done using the SPSS software (version 17.0, SPSS Inc., Chicago, IL, USA).

## 3. Results

### 3.1. Morphological Examination and Particle-Size Measurement

#### SEM and EDS Analysis

The four nanocolloidal samples TCA-AgNP, TCA-AgNP-PI, Cinn-AgNP, and Cinn-AgNP-PI were investigated morphologically by SEM (Figure 1). Spherical silver particles are formed due to the aggregation of small particles with high surface free energy. In both cases of chemical and green reduction of AgNO_3_, the resulting nanoparticles were small in size, indicating efficient synthesis. All particles were reported in a size range between 30–70 nm. The nanoparticles were almost spherical in shape and polydispersed in nature. The EDS of each corresponding sample is shown in Figure 1. The EDS shows the presence of silver and carbon in all samples while iodine appears only in the two samples TCA-AgNP-PI and Cinn-AgNP-PI after the addition of povidone iodine. The percentage of silver was quantified by EDS analysis for TCA-AgNPs, TCA-AgNPs-PI, Cinn-AgNPs, and Cinn-AgNPs-PI as 0.2, 0.5, 2.3, and 2.6%, respectively. The morphology and distribution of AgNPs deposited on the surgical sutures are demonstrated by SEM in Figure 2. The element Al appears in all measurements because of using an aluminum holder during the measurements. The plain suture and coated suture can be compared through SEM as well. SEM analysis exposed the typical interweaved nature of the multifilamented, absorbable suture and the operative deposition of the nanocomposites on the surface of the treated sutures. All the formulations developed were homogeneous and stable over long periods (at room temperature and in the dark).

The four nanocomplexes TCA-AgNP, TCA-AgNP-PI, Cinn-AgNP, and Cinn-AgNP-PI were investigated morphologically by EDS layered images (Figure 3). The images confirm the uniform distribution of the components in the samples (Figure 3).

### 3.2. Particle Size, PDL, and Zeta Potential of TCA-AgNP, TCA-AgNP-PI, Cinn-AgNP, and Cinn-AgNP-PI

#### 3.2.1. Dynamic Light Scattering (DLS)

The sizes of the nanocolloidal samples are ranged between 50 and 100 nm. Table 1 illustrates the results acquired for the measurements of four samples. The average size is below 100 nm. The polydispersion values are needed to characterize nanoparticles. The low values in our four samples are a proof of the uniformity and quality of the prepared dispersions. Figure 4 represents the size distribution of nanocomposites by intensity. The DLS data exhibits a unimodal size distribution already confirmed by SEM measurements (Figure 1, Figure 2 and Figure 3). There is a prevalence of medium-sized nanoparticles. No other peaks of intensity are detected (Figure 4). This confirms the absence of bulky aggregates and a broad size distribution as reported in previous papers [74,75]. The reducing agents TCA and Cinn controlled the size distribution and size-directing properties of the Cinn bark extract is confirmed [75]. 

#### 3.2.2. Zeta Potential Analysis

Zeta (ζ)-potential measurements provide significant data about agglomeration, distribution and stability of nanoparticles in the solution. The AgNPs samples have ζ-potential values close to −30 mV (Table 1 and Figure 5). The negative charge is a crucial factor for the negatively charged ions or compounds due to the presence of TCA, metabolites from cinnamon and iodine. These compounds have a high negative charge at neutral pH. 

The sample TCA-AgNP has a moderate charge up to −17 mV (Table 1). Considering the sizes obtained and their ζ-potential, the sample results were stable and did not form aggregates as verified by the analyses. Table 1 shows the particle-size distribution and the data indicates, that all the prepared AgNPs were polydisperse, as it was specified by the PDI values. This could be related to variations in growth rates of individual particles in the nucleation step.

### 3.3. UV-Vis Spectroscopic Analysis of TCA-AgNP, TCA-AgNP-PI, Cinn-AgNP, and Cinn-AgNP-PI

The synthesis of AgNPs was verified by measuring the absorption spectra of synthesized nanoparticles against particular capping agent. The absorption of the AgNPs is observed near 410–430 nm in the UV–vis spectrum for all samples, which is due to surface plasmon resonance of conducting electrons from the surface AgNPs (Figure 6). Additional peaks were seen in the spectrum, confirming the presence of different capping agents TCA, Cinn, and PI in our four nanocolloidal samples. 

The UV-vis spectra indicate lower wavelengths before the addition of PI. The wavelength is directly correlated to the size of nanoparticles. After adding PI, the wavelength shifted to a higher value due to increase in the size of the silver nanoparticles. Figure 6a indicates the effect of PI addition to TCA-AgNPs. The peaks near 290 and 250 nm in both figures are related to PI. Figure 6b shows the characteristic and sharp peak for AgNPs indicating the narrow range of the size distribution of AgNPs synthetized via green method. These results are in agreement with the DLS data. The spectra confirm the reduction proficiency of the cinnamon bark extract as well (Figure 6b).

### 3.4. FT-IR Spectroscopic Analysis of TCA-AgNP, TCA-AgNP-PI, Cinn-AgNP, and Cinn-AgNP-PI

FTIR measurements were used to investigate the possible organic molecules structures in each sample. Figure 7 elucidates the FTIR spectra of the prepared AgNPs with TCA and cinnamon bark extract before and after the addition of PI. 

The spectra show mutual peaks due to common groups in the four nanocolloidal samples. The broad bands at 3410.74, 3297.88, and 3016.53 cm^−1^ (Figure 7b–d) are related to NH (amide group) and OH (alcohol group) stretching. These bands can be found in plant compounds and PI as well. The peaks at 2936.01 and 2871.81 cm^−1^ are anti-symmetric and symmetric stretching of CH_2_, respectively (Figure 7a,b). The peaks for the carbonyl- and carboxylic groups are available in all figures around 1732.08, 1626, 1610, and 1430 cm^−1^. 

The electrostatic interaction between the positive charge of naked AgNPs and π-electrons in the carbonyl groups of organic compounds in the cinnamon extract provide an explanation for the absorption of these molecules onto the metal nanoparticles [74]. These carbonyl groups that are attached to the surface of metal nano-scaled particles in the Cinn-AgNP and Cinn-AgNP-PI are related to flavonones or terpenoids originating from the natural cinnamon bark extract. In the nanocolloids TCA-AgNP and TCA-AgNP-PI these peaks are associated with the carboxylic group in TCA. The organic groups had stronger capacity to bind with metal nanoparticles, acting as capping and stabilizing agents. The peaks at 1031, 1228, and 1101 cm^−1^ (Figure 7b–d) were due to the C–O stretching vibrations of polyols, ethers, and alcoholic groups, respectively. The peaks around 1019 cm^−1^ indicate the C-OH stretching vibration on the AgNP surface as reported by Khan et al. [78]. Peaks at 528.55 and 760 cm^−1^ are corresponding to alkene bending and carboxylic group stretching, respectively (Figure 7a–d). 

### 3.5. X-Ray Diffraction Measurement of TCA-AgNP, TCA-AgNP-PI, Cinn-AgNP, and Cinn-AgNP-PI

Figure 8 illustrates the XRD patterns of TCA-AgNPs-PI and Cinn-AgNPs-PI synthesized using TCA and cinnamon bark extract, then coated with povidone iodine polymer. The XRD patterns of these two samples indicate a similar diffraction profile. The XRD graphs accordingly demonstrate that the AgNPs formed in this study were not completely crystalline in nature. The XRD pattern proved that the main crystalline phase was silver, and there were no apparent other phases as impurities.

The XRD pattern of the pure PI indicates the presence of crystalline behavior in agreement with results reported earlier [79]. The presence of AgNP disturbs the semicrystalline nature of PI. Furthermore, the XRD pattern confirmed that the prepared silver nanocomplex is nearly amorphous because crystalline peaks can be seen over the entire range of 2θ degrees. Obviously, only the hump from 2θ *=* 59° to 2θ *=* 67° can be observed. Other researchers observed such humps for organometallic materials [80]. Therefore, XRD analysis results authenticate that reaction takes place between the PI polymer and the synthesized organosilver material. Study of the optical band gap is necessary to analyze the fact that PI polymer in this complex is almost amorphous. Figure 8 shows the broad peak for (220) of the Ag metal structure. Figure 8b indicates the presence of Ag metal and AgCl structure in the compound Cinn-AgNP-PI. Al Aboody confirms also the presence of an AgCl secondary phase in AgNPs samples synthesized using plant extract [81].

### 3.6. SERS Analysis of TCA-AgNP, TCA-AgNP-PI, Cinn-AgNP, and Cinn-AgNP-PI 

Figure 9 shows the results of the four samples. The last sample (Cinn-AgNP-PI) is Surface-Enhanced Raman Spectroscopy (SERS)-inactive. 

Raman enhancement is clearly observed in Figure 9 for TCA-AgNP, CA-AgNP-PI, and Cinn-AgNP. Raman spectroscopy was included in order to investigate the potential functional groups of green and chemical capping agents, which were used in this study to stabilize silver nanoparticles. In Figure 9a–c, the peaks at 1052 and 1210 cm^−1^ and 900 cm^−1^ are related to the C–H in-plane bending and out-of-plane vibration due to the organic structure in TCA, cinnamon bark extract and PI, respectively. The broad peaks around 1650 cm^−1^ are symmetric and asymmetric C=O stretching vibrations of carboxylate group- and C-N amide group vibrations, respectively, in PI molecules (Figure 9b) and cinnamon bark extract biomolecules (Figure 9c) [82]. Cinn-AgNP-PI is SERS-inactive (Figure 9d).

A sharp peak at 1000 cm^−1^ reveals the stretching vibrations of Ag–O from the carbonyl C=O groups in TCA and Cinn (Figure 9a–c) [83]. These peaks appear also in the FT-IR spectra and confirm the composition of our nanocolloids TCA-AgNP, CA-AgNP-PI, and Cinn-AgNP (Figure 7a,b) [78,82]. Figure 7d reveals a corresponding small peak and confirms stretching vibrations of Ag-O from carboxyl- and carbonyl groups in the cinnamon extract Cinn. These results indicate adsorption of carbonyl-oxygen groups in TCA-AgNP, TCA-AgNP-PI, Cinn-AgNP, and Cinn-AgNP-PI to the AgNP surface and are presented in a scheme in Figure 10. 

### 3.7. Antimicrobial Activities of TCA-AgNP, TCA-AgNP-PI, Cinn-AgNP, and Cinn-AgNP-PI 

#### 3.7.1. Determination of Antimicrobial Properties of TCA-AgNP, TCA-AgNP-PI, Cinn-AgNP, and Cinn-AgNP-PI

The four compounds TCA-AgNP, TCA-AgNP-PI, Cinn-AgNP, and Cinn-AgNP-PI were tested against 3 Gram-positive (*S. pneumoniae ATCC 49619, S. aureus ATCC 25923*, *E. faecalis ATCC 29212*), 2 Gram-negative (*E. coli WDCM 00013*, *P. aeruginosa WDCM 00026*) bacterial reference strains and the fungus *C. albicans WDCM 00054* by agar-well and disc dilution assays (Table 2). The three common antibiotics gentamycin (G), cefotaxime (CTX), and nystatin (NY) served as positive controls (Table 2). The disc diffusion studies were performed in dilution series of 50 µg/mL (D), 25 µg/mL, and 12.5 µg/mL (D*) TCA-AgNP, TCA-AgNP-PI, Cinn-AgNP, and Cinn-AgNP-PI. At the concentration of 25 µg/mL there are no significant changes in the dilution series and the results are within the expected ranges. The negative controls methanol and water showed no zone of inhibition (ZOI). All these results were not explicitly added to Table 2.

The compound TCA-AgNP-PI inhibited Gram-negative pathogens stronger than Gram-positive species and *C. albicans*. Disc diffusion studies resulted in smaller ZOI compared to agar well diffusion tests (Table 2). TCA-AgNP-PI exhibit the highest inhibition zone (27 mm) in *P. aeruginosa WDCM 00026* followed by *S. aureus ATCC 25923* (ZOI = 20) both at concentrations of 50 µg/mL (Table 2, Figure 11). TCA-AgNP-PI had intermediate antifungal activity towards *C. albicans* (ZOI = 13). Cinn-AgNP and Cinn-AgNP-PI had a weaker effect on microbial growth control than TCA-AgNP-PI on the same microorganisms (Table 2).

#### 3.7.2. Determination of Antimicrobial Properties of Sutures Dip-Coated with TCA-AgNP, TCA-AgNP-PI, Cinn-AgNP, and Cinn-AgNP-PI

The four compounds TCA-AgNP, TCA-AgNP-PI, Cinn-AgNP, and Cinn-AgNP-PI were used to dip-coat PGA sutures. The PGA sutures (2.5 cm) were treated with acetone, dried, and impregnated with the compounds for 18 h. After drying the sutures at ambient temperature, they were subjected to diffusion assay on 5 Gram-positive (*S. pneumoniae ATCC 49619, S. aureus ATCC 25923, E. faecalis ATCC 29212, S. pyogenes ATCC 19615, B. subtilis*), 4 Gram-negative (*E. coli WDCM 00013*, *P. aeruginosa WDCM 00026, K. pneumoniae WDCM 00097, P. mirabilis ATCC 29906*) bacterial reference strains and the fungus *C. albicans WDCM 00054* (Table 3). The three common antibiotics gentamycin (G), cefotaxime (CTX), and nystatin (NY) served as positive controls (Table 3). The diffusion results of PGA sutures dip-coated with 50 µg/mL (S) of each TCA-AgNP, TCA-AgNP-PI, Cinn-AgNP, and Cinn-AgNP-PI are listed in Table 3.

Dip-coated sutures with TCA-AgNP-PI and TCA-AgNP inhibited bacterial pathogens stronger than both Cinn compounds (Figure 12, Figure 13 and Figure 14, Table 3). The selected Gram-negative species were more susceptible to TCA-AgNP-PI and TCA-AgNP than the selected Gram-positive bacteria. *K. pneumoniae WDCM 00097* was inhibited strongly by three of our compounds (Figure 12) followed by *E. coli WDCM 00013* and *P. aeruginosa WDCM 00026. S. aureus ATCC 25923* had the highest susceptibility towards TCA-AgNP-PI (Table 3, Figure 13 and Figure 14). All four nanocolloid impregnated sutures failed to show antifungal properties against *C. albicans* (Table 3).

## 4. Discussion

Antibiotic resistance is a serious problem for mankind and is one reason for increasing morbidity and mortality rates [1,2]. Plant extracts are in the spotlight as a promising solution because they consist of many bioactive compounds and pharmacokinetic synergists, which can increase the pharmaceutical value of drugs [73]. Trans-cinnamic acid (TCA) is one of the active constituents in cinnamon bark and has antimicrobial activity [84]. Many groups investigated biocidal properties of cinnamon bark crude extract (Cinn) [64,65,85]. Wound care products and surgical sutures are known for their affinity to microbial growth and biofilm formation on their surfaces [86,87]. Coating PGA sutures with antimicrobial nano-materials derived from TCA and Cinn may aid wound closure, reduce time and costs spent for health care. Iodine can play a positive role as microbicidal coating but disadvantages like skin irritation, discoloration and short durability by uncontrolled free iodine release are drawbacks [26,30]. The commonly used disinfectant povidone iodine is reported to have similar adverse effects and incorporates triiodide ions for the release of free iodine [48,49,50]. Silver nanoparticles have antimicrobial effects and are incorporated increasingly in many products, including surgical sutures [58,59]. Sustainability and cytotoxicity can be minimized by reducing AgNP content through combinations with other antimicrobials. In this regard, we investigated antimicrobial compounds consisting of AgNP, PI and bioactive compounds derived from *cinnamomum zeylanicum*. In a previous report, absorbable sutures were coated with curcumin loaded gold nanocomposites [88]. The results were promising regarding antibacterial functionality and biocompatibility [88]. We present a novel silver nanocomposite that can be used as coating for PGA absorbable surgical sutures enhancing its antimicrobial properties. TCA-AgNP, TCA-AgNP-PI, Cinn-AgNP, and Cinn-AgNP-PI were developed to maximize microbial growth control. The nanocolloids and their dip-coated sutures were characterized morphologically and tested for their in-vitro antimicrobial activities. Further investigations included FT-IR, UV-Vis, XRD, Zeta potential, and Raman measurements. The sutures were treated with our four samples in aqueous solution with the concentration of 50 µg/mL of each, via the dip-coating deposition technique. 

The analytical data of our compounds TCA-AgNP, TCA-AgNP-PI, Cinn-AgNP, and Cinn-AgNP-PI confirm spherical shape and smaller size compared to curcumin loaded gold nanocomposites [88]. EDX analysis of our samples are shown in Figure 1. Silver, iodine, and organic materials are present and confirm TCA, Cinn, and PI as capping agents of AgNP. The AgNPs are usually surrounded by positive charge due to unreduced silver ions. Capping agents stabilize AgNP by preventing agglomeration and oxidation (from silver to silver ions). The capping agents TCA, Cinn, and PI are adsorbed on the surface of AgNP through their partly negatively charged oxygen atoms in their carbonyl groups [74]. This renders a negative charge on the AgNP surface. At the same time, the organic polymer PI acts as a surfactant like PVP, preventing agglomeration through steric crowding and repulsion. The stabilizing effect of the utilized capping agents TCA, Cinn, and PI is therefore confirmed. 

The hydrodynamic size and surface charge of the four nanocomposites were measured by DLS and ζ-potential (Figure 4 and Figure 5; Table 1). Zeta potential measures the charge of nanoparticles and their electrostatic repulsion. Our complex nanoparticles have values around −30 mV (Table 1). The negative charges from the ionized TCA, the polyphenols in the Cinn and the PI are predominant in the solution. The ζ-potential values of AgNP dispersions (Table 1) proved the efficiency of the capping agents in stabilizing the NPs by providing intensive negative charges. This result is in agreement with previous investigations using cranberry powder aqueous extract [89]. In our compounds the ζ-potential values change with the introduction of PI to higher negative zeta potentials (Table 1). The triiodide ions within the PI are the reason for this increased negative charge [50]. TCA-AgNP has a zeta potential of −17.9 mV and is reduced to −33.3 mV in TCA-AgNP-PI. The zeta-potential value of Cinn-AgNP slightly changes from −32.9 mV to −33.1 mV when PI is added (Table 1). Generally, TCA-AgNP has the highest zeta potential with −17.9 MV compared to the other compounds which show values around −33 mV. The mean particle sizes reduce when PI is added to TCA-AgNP and Cinn-AgNP from 61.3 nm to 57.2 nm and 88.1 nm to 51.5 nm, respectively. Cinn-AgNP-PI has the smallest mean particle size compared to the other three compounds. Adding PI into the Cinn-AgNP reduced the particle size dramatically. Cinn alone, with its bioactive compounds, did not stabilize and cap the AgNP as strong as the TCA to prevent agglomeration. The phenolic compounds in Cinn did not exhibit a strong synergistic effect in capping and stabilizing the silver nanoparticles. After adding the PI, the mean nanoparticle particle size decreases. As a result, PI is a better stabilizing and capping agent than Cinn extract. 

TCA-AgNP exhibits a smaller mean particle size than Cinn-AgNP. Therefore, TCA is a stronger capping and stabilizing agent than Cinn. TCA coats the nanoparticles surface homogenously and adds a hydrophobic nature to the nanoparticles with its hydrophobic benzene rings exposed towards the solution. In this context, the steric hindrance and repulsion between the lipophilic benzene rings prevent nanoparticles agglomeration. Furthermore, these hydrophobic nanoparticles may tend to move toward the interface and leave their bulk colloids due to the Brown’s motion effect [90]. The polyphenols in the crude extract of cinnamon have stabilizing and capping effects [91]. Our aqueous extract Cinn renders negative charges and steric hindrance as well but cannot prevent agglomeration as effectively as the TCA and PI. Cinn consists of many water-soluble phenolic compounds with different properties. The use of the Cinn plant extract of *cinnamomum zeylanicum* did not render the expected positive synergistic effects.

Figure 2 displays the homogeneity and adherence of the nanocomplex on the multi-filament absorbable suture. Previous studies have produced stable and adherent coatings on cotton using AgNPs biologically synthesized with leaf biomolecules of *C. roxburghii* [92]. We followed a coating and drying process of 18 h proposed previously to optimize the stability and effectiveness of the products [87]. In this work, the physical adsorption phenomena can be adopted to explain the bonding between the nanocomposites and the PGA suture material. The surface attachment according to this phenomenon happens commonly between modified silver nanocomposites and the target surface like sutures [93]. Physical adsorption includes Van der Waals forces for the adsorption of nanocolloids on the surface of an adsorbent [93]. Electrostatic forces and hydrogen bonding can lead also to physical adsorption [93]. PGA is a negatively polarized polymer with many electronegative oxygen atoms. TCA, phenolic compounds in Cinn and PI are adsorbed by their oxygen atoms in their C=O groups on the silver metal surface. The rest of the molecules consist of benzene groups (TCA/Cinn) and alkyl-chains (PI). These groups are adsorbed by dipolar attraction to the polar TGA polymer. Hydrogen bonding occurs between the hydrogen atoms of the H-O- groups within the carboxyl entity in TCA and the oxygen atoms of the carbonyl and ether groups in PGA. Cinnamaldehyde is the major constituent of *cinnamomum zeylanicum* aqueous extract [63]. Like TCA, cinnamaldehyde will be physically adsorbed by hydrogen bonding between the hydrogen atoms of the carbonyl group in cinnamaldehyde and the oxygen atoms of carbonyl and ether groups in PGA.

UV–visible spectroscopy is used to observe size and formation of nanoparticles in aqueous dispersions. The reduction of silver was elucidated via UV-vis spectroscopy with a plasmonic peak over the range of 400–450 nm. This is due to surface plasmon resonance phenomenon of the electrons in the conduction band of silver. The UV-vis spectrometric analysis of the final reaction mixtures TCA-AgNP, TCA-AgNP-PI, Cinn-AgNP, and Cinn-AgNP-PI showed absorption peaks around 390–415 nm, which are the characteristic peaks for AgNPs (Figure 5a) in agreement with previous studies with values around 412 nm [64,94]. Soni et al. synthesized AgNP with *cinnamomum zeylanicum* and reported a broad peak at 480 nm [66]. The UV-vis spectra of Cinn-AgNPs and Cinn-AgNP-PI show a distinct peak for AgNP at 405 nm which is an indication of reduction of silver (Figure 5b). The sharper peak in case of TCA-AgNP reveals a faster reduction rate for TCA compared to the green method with Cinn extract. There are new peaks characteristic for PI at around 244 and 288 nm after adding the PI [95]. TCA alone has a maximum absorbance peak at 265 nm in the spectrum, while TCA-AgNP gave a characteristic surface plasmon resonance band for AgNPs around 400 nm and proves effective stabilization of AgNPs by TCA. The addition of PI to TCA-AgNP shifted the absorption peak from 390–402 nm to 410–415 nm at the same absorbance of 0.56. This bathochromic shift denotes an absorption change to lower energy with the addition of PI. The red shift to longer wavelengths is due to inner-particle interactions between TCA and PI [96]. The hydrophobic TCA with its benzene rings towards the surroundings of the nanocomposite exhibits repulsion contributions to frequency shifts when hydrophilic PI is added to the surface of the AgNP.

TCA-AgNP and TCA-AgNP-PI absorb both at 0.56, Cinn-AgNP and Cinn-AgNP-PI at 0.86 and 0.92, respectively. Absorbance of light is inversely related to the density of surrounding molecules adsorbed on the AgNP surface. The higher the absorbance in the UV-vis spectrum, the less molecules are adsorbed on the AgNP surface. This indicates that the AgNP surface in TCA-AgNP and TCA-AgNP-PI is more coated and populated by adsorbed TCA and PI than in Cinn-AgNP and Cinn-AgNP-PI. TCA and PI are having better capping and stabilizing properties than Cinn and PI together. Adding PI into Cinn-AgNP reduces the overall coating of the AgNP and increases the absorbance from 0.86 to 0.92. These results are due to the mixed composition of the Cinn sample consisting of different biomolecules. The phenolic compounds within the Cinn extract with their hydrophobic benzene rings lead to steric hindrance, crowding, and repulsion. After adding the hydrophilic polymer PI, the repulsion, crowding, and steric hindrance increase, resulting in desorption of different phenolic compounds. At the same time, this process leads to stronger encapsulation of AgNP by the remaining phenolic compounds and PI. All these observations were also confirmed in the previous parts regarding mean particle size and zeta potential. Again, TCA remains a better stabilizing and capping agent than Cinn, preventing agglomeration and forming smaller sized NPs. 

FT-IR analysis was conducted in order to identify the nature of various organic functional groups in biomolecules responsible for the reduction and capping of AgNPs. FT-IR analysis of pure TCA, PI and Cinn (Appendix A) compared with the spectra of the nanocolloids confirmed the composition of TCA-AgNP, TCA-AgNP-PI, Cinn-AgNP, and Cinn-AgNP-PI (Figure 7). The FT-IR spectra of our four products are showing evidence of the presence of organic material surrounding the silver nanoparticles (Figure 7). The peaks for C-OH stretching vibrations on the AgNP surface are available around 1019 cm^−1^ with different strength and sharpness in TCA-AgNP, TCA-AgNP-PI, Cinn-AgNP, and Cinn-AgNP-PI (Figure 7a–d) [78]. Bands with small intensity around 1340–1400 cm^−1^ and 1640 cm^−1^ reveal AgNP-carboxylate interactions (Figure 7a–d) [82,97].

Figure 7a reveals the characteristic ring-stretching peak at 1540 cm^−1^ verifying the satisfactory stabilizing effect of the TCA surrounding the AgNPs. Another peak for the C=O bond stretching of the carbonyl group in TCA is available at 1690 cm^−1^. After adding PI to the TCA-AgNP, this peak increases in intensity, while another sharp peak at 1540 cm^−1^ appears. These are attributed to C=O and C-N respectively (Figure 7b). These two active groups are related to PI. The addition of PI into TCA-AgNP increases the number of C=O interactions with the AgNP surface originating from TCA and from PI. PI molecules did not replace TCA molecules on the AgNP surface, therefore the intensity of the peak for C=O bond stretching increases. A strong peak bonding at peak 3420 cm^−1^ represents the (N-H) bond stretching in the amine groups and 2920 cm^−1^ corresponds to C–H bond stretches in alkene groups (Figure 7b) [98]. The same peaks appear after adding PI into Cinn-AgNP in Figure 7d. Prekumar et al. investigated biosynthesized silver nanoparticles by using *cinnamomum zeylanicum* and reported in their respective FT-IR analysis a peak at 3259 cm^−1^ [64]. They attribute this signal to the vibrations of the O-H bond in the carboxylic acid [64]. It is missing in the spectrum of TCA-AgNP due to the presence of carboxylate ions. Accordingly, the same sharp peak at 3378 cm^−1^ in the nanocolloidal sample Cinn-AgNP corresponds to the O-H bond stretching in hydroxyl groups found in polyphenols as reported by Premkumar et al. (Figure 7c) [64]. This signal can also be related to the formation of carboxyl groups, when cinnamaldehyde reduces the silver and is oxidized to cinnamic acid. According to Doyle et al. *cinnamomum zeylanicum* extract consists of 90.5% cinnamaldehyde [63]. The peak at 1730 cm^−1^ is for the C=O bond stretching attributed to the biomolecules naturally found in crude cinnamon extract and therefore is related to the carbonyl group in the cinnamaldehyde itself. AgNP adsorbs the carbonyl-oxygen group on its surface as stated above [74]. Figure 7c, for the cinnamon extract capped AgNPs after addition of PI, elevates the presence of the characteristic peaks of PI as described in Figure 7b. The peak for the alcohol group decreases due to the addition of PI. This indicates that part of the cinnamic acid molecules and further biomolecules available in the Cinn extract are replaced by PI molecules. The peak for the C=O stretching remains the same after adding PI. As a result, PI does not replace the cinnamaldehyde, cinnamic acid and biomolecules completely. Crowding and repulsion lead to exchange of some phenolic compounds with PI in small scale.

XRD analysis was done for the two compounds TCA-AgNPs-PI and Cinn-AgNPs-PI to identify the crystallinity of the samples. The XRD spectra of the TCA-AgNPs-PI sample contain four distinctive peaks at 2θ = 44.4°, 64.5°, and 81.5° (weak) that are related to the crystalline nature of the reduced AgNP (Figure 8a). These correspond to metallic silver with planes of face-centered-cubic structure (200), (220) and (311), respectively [82,99,100]. Premkumar et al. reported peaks at 27.74°, 32.16°, 38.03°, 46.16°, and 54.72° for a size range of 40–70 nm [64]. The peaks in our samples are broad due to the narrow range of the size distribution (52–57 nm). The crystallite size was calculated from the silver width at half maximum by using Debye–Scherrer equation, and was found to be around 27 nm [95,98]:*D* = *C*λ/β*cos*θ(1)
where *D* is crystal (particle) size (A°), *C* is the shape factor that was equal to 0.94, λ is the wavelength of X-ray used (1.54060 A°), β is full-width at half-maximum (FWHM) in radians, and θ is Bragg’s diffraction angle for the peak in degrees. The grain size of AgNPs is accordingly in the range of 26 nm. The X-ray diffraction of PI shows two peaks close to 2θ equals to 11.68°θ and 23.84°θ due to the amorphous nature of PI (Figure 8a) [101,102]. The X-ray pattern for the sample Cinn-AgNPs-PI shows a maximum intensity peak at 23°, following maximum peak at 64° and a less intense peak at 81.5° (Figure 8b). The first peak is related to the PI and its amorphous nature. The following two peaks are attributed to metallic AgNP with planes of face-centered-cubic structure (220) and (311), respectively. These findings confirm the semicrystalline nature of the silver nanoparticles. Additionally, there is very weak peak for AgCl at 2θ = 27.4° for (200), which is reported by Kumar et al. and Hamed et al. as well [99,100]. The XRD analysis of the nanocompounds TCA-AgNPs-PI and Cinn-AgNPs-PI indicated both amorphous and crystalline regions [101].

Surfaced-enhanced Raman scattering (SERS) is a method used to characterize the nature of chemical bonds formed on the surface of the metal and the molecular conformation on the specific metal films [102,103,104,105]. In SERS measurements, molecular vibrations perpendicular to the surface should be greater in the spectra, while those parallel to the surface should be deteriorated [102,103,104,105]. The SERS spectra of the compounds TCA-AgNP, TCA-AgNP-PI, Cinn-AgNP, and Cinn-AgNP-PI are shown in Figure 9. 

The peak at 1605 cm^−1^ can be assigned to the stretching vibration of C=O, indicating that TCA, cinnamon bark biomolecules, and PI molecules interact with the AgNP surface mostly through their oxygen in their carbonyl group C=O [82,97]. This was also confirmed by UV-vis and FT-IR analysis above. Figure 9b,c show a band at 1346 cm^−1^ due to the stretching vibration of C–N and the bending vibration of C–H. The significant enhancement of the signal at 1000 cm^−1^ (due to ring breathing and asymmetric stretching vibration of CH_2_ in the skeletal chain of PI, TCA, and biomolecules from plant extract) reveals that the CH_2_ chain is close to the surface of AgNPs. The results of our SERS analysis are in agreement with previous reports and discussions above [102,103,104,105,106]. The nanocolloidal sample Cinn-AgNP-PI is SERS-inactive (Figure 9d). SERS signals are recorded, if our spherical, monodispersed AgNP aggregate and induce plasmon coupling between the NPs resulting in hot spots [107]. SERS signals are not detected in Cinn-AgNP-PI because NP clusters did not form [107]. Cinn-AgNP-PI has the smallest particle size compared to our other compounds TCA-AgNP, TCA-AgNP-PI, and Cinn-AgNP. When PI is added to the Cinn-AgNP, biomolecules with less physical adsorption to the silver metal surface are replaced partly by the polymeric PI as discussed above. The capping agent PI reduces the agglomeration, mean particle size decreases dramatically (to 51.5 nm, Table 1), plasmon coupling between the AgNPs are reduced, and hot spots cannot be formed to achieve SERS enhancement. This mechanism renders Cinn-AgNP-PI to be SERS-inactive (Figure 9d).

The antimicrobial activities of our four samples TCA-AgNP, TCA-AgNP-PI, Cinn-AgNP, Cinn-AgNP-PI and their dip-coated PGA sutures were tested against 6 pathogens compared to 3 common antibiotics (cefotaxime, gentamycin and nystatin). Gram-positive cocci included *Streptococcus pneumoniae (S. pneumoniae ATCC 49619)*, *Staphylococcus aureus (S. aureus ATCC 25923)*, *and Enterococcus faecalis* (*E. faecalis ATCC 29212).* Gram-negative bacteria were *Pseudomonas aeruginosa (P. aeruginosa WDCM 00026)* and *Escherichia coli (E. coli WDCM 00013).* The tested fungus was *Candida albicans WDCM 00054. E.coli*, *S. aureus*, P. *aeruginosa*, *S. pyogenes*, and *K. pneumoniae* are present in wounds and belong to the group of ESKAPEE pathogens [1,2,3]. 

According to our viewpoint, the hybrid complex of AgNP, PI, and bioactive compounds can be a sustainable combination of known antimicrobial agents with increased biocompatibility, microbial growth control, and minimized adverse effects of PI and AgNP [49,60]. 

Previous investigations report the biocidal properties of cinnamon bark crude extract (Cinn) [64,65,82]. Cinnamaldehyde is slightly soluble in water, but at the same time it is one of the major constituents in the bark of *cinnamomum zeylanicum* [63]. The extraction in water leads to a high amount of cinnamaldehyde and other plant constituents [63]. These reduce the silver ions in the silver nitrate solution to silver nanoparticles in the Cinn-AgNP sample. Cinnamaldehyde is oxidized to cinnamic acid and is available in the FT-IR spectra (Figure 7c). 

The pH of the reaction solution was maintained at pH 5. In this acidic environment, the phenolic acids do not dissociate their protons and therefore remain lipophilic [67]. This allows the cinnamic acid molecules to enter the lipophilic cytoplasmic membrane and exert in this way its antimicrobial activity on the microbial cells. The electrophilic β-carbon atom in the *trans*-cinnamic acid molecule establishes electrostatic interactions with the negatively charged cell membranes of the microorganisms. These interactions may lead to adsorption, deformation of the membranes and their destruction. Another action of TCA is oxidizing the AgNPs. This leads to the release of silver ions, which are detrimental for the cell membranes. 

Another antimicrobial action of *trans*-cinnamic acid is due to its phenolic ring, which interacts with the proteins in the cytoplasmic membranes and causes leakage of cell constituents [70]. TCA can be used as a potential food preservative to inhibit the bacterial growth [70,84]. 

Povidone iodine has microbicidal activity due to release of free molecular iodine. The hydrophilic PI is in aqueous environment not stable and this reduces its antimicrobial properties and durability. We stabilized PI with the hydrophobic component TCA, which also acted as encapsulation and reducing agent for AgNP. 

By incorporating PI into our AgNP-nano-carrier system we targeted synergistic antimicrobial effects based on the three main components. AgNP adsorbs TCA, the bioactive compounds originating from the cinnamon bark extract and PI molecules through interactions with their oxygen atoms in the C=O groups [74,78]. The microbial growth control mechanisms of PI are induced by electrostatic interaction with microbial cell membranes triggering the release of free iodine from the triiodide ions [42,49]. The structural deformation of the nano-carrier system culminates in the release of all the components (AgNP, TCA, bioactive components in cinnamon bark extract), which in turn exert their individual antimicrobial activities against the tested pathogens. This nano complex was dip-coated on sutures to investigate their potential to prevent SSI. 

Our compounds TCA-AgNP, TCA-AgNP-PI, Cinn-AgNP, and Cinn-AgNP-PI were tested by agar well and disc diffusion methods against different pathogens and antibiotics. In the agar well dilution method, the wells were filled with 72 µL of our compounds with the concentration of 50 µg/mL (Table 2). The discs for the disc diffusion series were loaded each with 2 mL of 50 µg/mL, 25 µg/mL and of 12.5 µg/mL of compound, dried under ambient conditions and placed on the MHA agar. Agar well diffusion assay of the tested pathogens showed higher inhibition zones than the disc diffusion series of the same concentration (Table 2). In agar well series, TCA-AgNP-PI exerted the highest antimicrobial activity against the microorganisms compared to TCA-AgNP, Cinn-AgNP, and Cinn-AgNP-PI. TCA-AgNP-PI showed the highest inhibition towards *P. aeruginosa* with a (ZOI = 27 mm), followed by *S. aureus* (21 mm), *S. pneumoniae* / *E. coli* (17 mm) and *E. faecalis* (15 mm) (Table 2). TCA-AgNP had higher antifungal activity than TCA-AgNP-PI against *C. albicans* with 18 mm and 13 mm, respectively. *P. aeruginosa* is stronger inhibited by our compound (27 mm) than by cefotaxime (20 mm) (Table 2). The disc diffusion studies showed highest susceptibility for *P. aeruginosa* (20 mm), followed by *E.coli* (18 mm), *S. aureus* (15 mm), *S. pneumoniae* (13 mm), and *E. faecalis* (12 mm) towards TCA-AgNP-PI (Table 2). *P. aeruginosa* was similarly sensitive towards the control antibiotic cefotaxime (20 mm) and TCA-AgNP-PI (20 mm). Our compound TCA-AgNP-PI inhibits *P. aeruginosa* in disc diffusion studies with a concentration of 50 µg/mL almost as strong as cefotaxime (30 µg/disc). 

Cinn-AgNP-PI inhibited *S. aureus* with an inhibition zone of 20 mm, followed by *P. aeruginosa* (15 mm), *C. albicans* (15 mm), *E. coli* (14 mm), and *E. faecalis* (11 mm). Cinn-AgNP inhibited *P. aeruginosa* intermediately with 15 mm, followed by *E. coli* (14 mm), *S. aureus* (14 mm), and *E. faecalis* (10 mm) (Table 2).

In comparison, TCA-AgNP-PI controlled the microbial growth stronger than Cinn-AgNP-PI, followed by TCA-AgNP and lastly Cinn-AgNP (Table 2). Previous reports agreed upon higher microbial growth control when Cinn-AgNP is compared to AgNP alone and TCA-nanoemulsion is compared to pure TCA [64,70]. The addition of PI is increasing the antimicrobial efficacy in our investigations. Malheiro et al. showed that cinnamaldehyde has stronger antimicrobial activity than cinnamic acid and its derivatives [67]. In our investigations, TCA in combination with AgNP is exerting higher antimicrobial effects than Cinn-AgNP (Table 2). *C. albicans* is resistant towards Cinn-AgNP, but Cinn-AgNP-PI has, in agar well diffusion methods, similar antifungal properties (15 mm) compared to the antibiotic nystatin (16 mm) induced by PI (Table 2). In the disc diffusion series, only TCA-AgNP exerts antifungal activity when diluted until 12.5 µg/mL. As a result, TCA-AgNP remains the strongest antifungal agent in our investigations, while TCA-AgNP-PI and Cinn-AgNP-PI only show antifungal activity in agar well studies with 13 mm and 15 mm, respectively (Table 2). The hydrophilic PI exerts its antifungal action in the well studies by diffusing through the MHA agar and releasing free iodine on *C. albicans*. Cinn-AgNP-PI is slightly stronger antifungal than TCA-AgNP-PI. *C. albicans* is resistant against Cinn-AgNP alone but is inhibited in agar well and disc diffusion studies by TCA-AgNP. The antifungal action originates from TCA. Cinn and AgNP have no antifungal activity even if they are combined in nanoparticles. 

In general, TCA-AgNP-PI inhibits Gram-negative bacteria strongly, followed by Gram-positive bacteria and finally *C. albicans* (Table 2). Cinn-AgNP-PI has higher effects on Gram-positive than Gram-negative bacteria. TCA-AgNP is mostly active against the Gram-negative pathogen *P. aeruginosa,* followed b *C. albicans* and further bacterial microorganisms (Table 2). Cinn-AgNP shows the highest activity against Gram negative *P. aeruginosa*, followed by Gram-negative *E. coli* and the Gram-positive pathogens *S. aureus* and *E. faecalis* only (Table 2). The Gram-negative, motile, and rod-shaped bacilli *P. aeruginosa* has the highest susceptibility towards three of our compounds TCA-AgNP, TCA-AgNP-PI and Cinn-AgNP in agar well diffusion studies. Cinn-AgNP-PI is more effective on *S. aureus* in agar well studies but shows similar activity against *P. aeruginosa* and *S. aureus* in disc diffusion studies. In general, Gram-negative bacteria (*P. aeruginosa and E. coli*) are slightly more susceptible than Gram-positive (*S. aureus* and *S. pneumoniae*) against our four compounds. As a result, our compounds are active against microorganisms reported as antibiotic resistant ESKAPEE pathogens, which can also form inter-kingdom biofilms in wounds leading to morbidity and mortality [1,2,3,20].

We dip-coated our compounds TCA-AgNP, TCA-AgNP-PI, Cinn-AgNP, and Cinn-AgNP-PI on biocompatible, braided PGA sutures and tested their antimicrobial activity against 10 different pathogens compared to 4 common antibiotics (chloramphenicol, cefotaxime, gentamycin, and nystatin) (Table 3). Gram-positive cocci included *Streptococcus pyogenes (S. pyogenes ATCC 19615)*, *Streptococcus pneumoniae (S. pneumoniae ATCC 49619)*, *Staphylococcus aureus (S. aureus ATCC 25923)*, *Enterococcus faecalis* (*E. faecalis ATCC 29212)*, and the spore-forming bacteria *Bacillus subtilis.* Gram-negative bacteria were *Pseudomonas aeruginosa (P. aeruginosa WDCM 00026)*, *Escherichia coli (E. coli WDCM 00013)*, *Klebsiella pneumoniae (K. pneumoniae WDCM 00097)*, and *Proteus mirabilis* (*P. mirabilis ATCC 29906).* The tested fungus was *Candida albicans, C. albicans WDCM 00054. E.coli*, *S. aureus*, *P. aeruginosa*, *S. pyogenes* and *K. pneumoniae* ESKAPEE pathogens and present in wounds [1,2,3].

Impregnated TCA-AgNP-PI had the highest antibacterial activity followed by TCA-AgNP, Cinn-AgNP and finally Cinn-AgNP-PI with the lowest activity (Table 3). TCA-AgNP-PI showed the inhibition zones of 5 mm against the Gram-negative *P. aeruginosa*, *E. coli*, *K. pneumoniae*, and Gram-positive *B. subtilis*, *S. aureus* (Table 3). In general, our dip-coated compounds inhibited Gram-negative species more than the Gram-positive (Table 3). TCA-AgNP showed the same results for the Gram-negative bacteria except for *K. pneumoniae* (4 mm) and the Gram-positive *B. subtilis* (4 mm). Cinn-AgNP have the highest antibacterial effects on the Gram-negative *K. pneumoniae* (5 mm), *P. aeruginosa* (3 mm), and the Gram-positive *S. pneumoniae* (3 mm) (Table 3). The later is resistant to all of the other three respective nanocomposites. *S. aureus* is susceptible towards TCA-AgNP-PI (5 mm) only.

Addition of PI into TCA-AgNP increases the bacterial growth control but does not make a difference for Cinn-AgNP and Cinn-AgNP-PI except in the case of *E. coli*. *C. albicans* and *E. faecalis* are resistant against the four compounds (Table 3). Our compounds lose their antifungal activity and their activity towards *E. faecalis* when they are impregnated on sutures. The complexes exhibited increased antifungal properties in agar well dilution after PI was introduced into the nanocolloid as explained above. This property is lost due to dip-coating on the suture. The sutures were coated, dried and placed on the MHA agar. The contact area of the suture on the MHA is smaller than in agar well and disc diffusion assays. The dryness of the suture is not helpful in disseminating the coating against the pathogens. In an in vivo experiment, the sutures are completely exposed from every side by the human tissues within a treated wound. This indicates increased antimicrobial activity of our compounds and support in preventing SSI and biofilm formation. 

The antimicrobial screening by agar well and disc diffusion methods of our pure complex nanocolloids and their impregnated sutures with TCA-AgNP, TCA-AgNP-PI, Cinn-AgNP, and Cinn-AgNP-PI have some results in common. Sutures and pure nanocomplexes exert higher inhibitory action against Gram-negative bacteria compared to Gram-positive. Staphylococci (*S. aureus*) are stronger inhibited than streptococci. Rod-shaped bacilli are more susceptible than round shaped cocci towards our compounds. Among bacilli, the susceptibility is inversely related to motility. The non-motile *K. pneumoniae* is more susceptible than motile species (*P. aeruginosa*, *E. coli*) and those with swarming motility (*P. mirabilis*). Among the Gram-positive cocci, the higher the complexity of the bacteria the higher the susceptibility. Clusters (*S. aureus*) are more susceptible than chains (*S. pyogenes*), short chains, pairs (*S. pneumoniae*), and single bacteria (*E. faecalis*). We reported similar trends previously for the triiodide complex with halogen bonding [42]. 

## 5. Conclusions

ESKAPEE pathogens are listed by WHO as immediate threat to our species due to their antibiotic resistance. Plant-based, biosynthesized silver nanoparticles, and povidone iodine may have future perspectives against pathogens, which develop resistance mechanisms against common drugs in a fast pace. Our target is to reduce the adverse effects of povidone iodine and silver nanoparticles by forming durable, biocompatible and sustainable nanocompounds. We investigated a combination of antimicrobial agents within nanocolloidal complexes. TCA-AgNP, TCA-AgNP-PI, Cinn-AgNP, and Cinn-AgNP-PI showed a narrow particle size distribution and proved to be microbicidal agents. Our compounds exerted antimicrobial activity on Gram-negative and Gram-positive pathogens at a concentration of 50 µg/mL as nanocolloidal samples and dip-coated sutures. TCA-AgNP, TCA-AgNP-PI, Cinn-AgNP, and Cinn-AgNP-PI and their dip-coated PGA sutures have potential for preventing SSI and biofilm formation in wound care. Future investigations should focus on in vivo application of our complexes and combinations with antibiotic drugs.

## Figures and Tables

**Figure 1 pharmaceutics-12-00361-f001:**
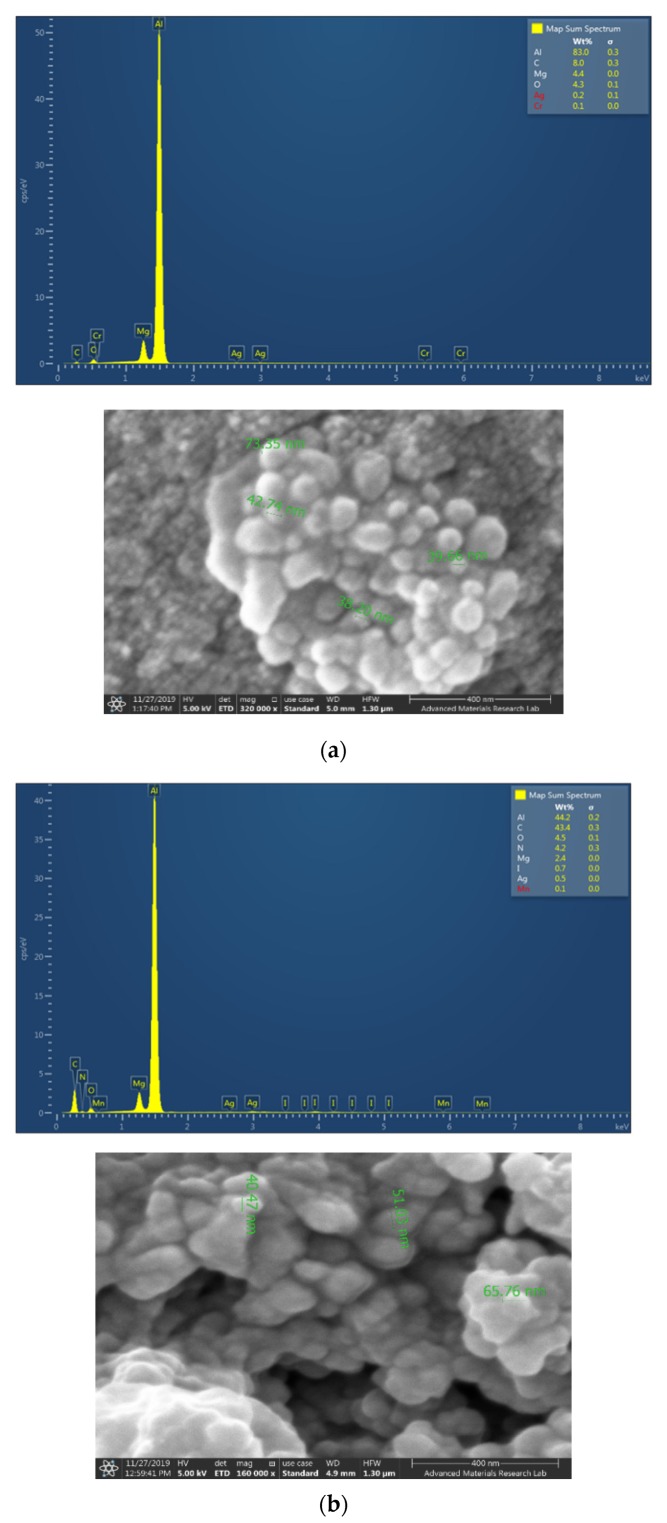
Energy dispersive spectroscopy (EDS) and corresponding scanning electron microscopy (SEM) of the four samples with *trans*-cinnamic acid (TCA), silver nanoparticles (AgNP), *cinnamomum zeylanicum* extract (Cinn) and povidone iodine (PI). From up to down: (**a**) TCA-AgNP; (**b**) Cinn-AgNP; (**c**) TCA-AgNP-PI; (**d**) Cinn-AgNP-PI.

**Figure 2 pharmaceutics-12-00361-f002:**
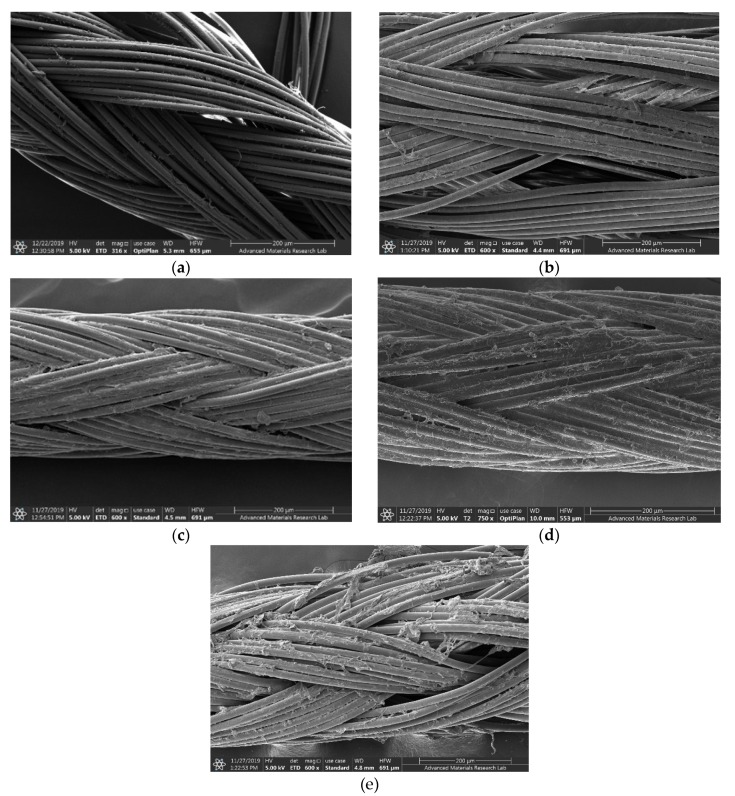
SEM of medical Polyglycolic Acid (PGA) sutures. From left to right: (**a**) Control (plain PGA suture); sutures coated with (**b**) TCA-AgNP; (**c**) Cinn-AgNP; (**d**) TCA-AgNP-PI; (**e**) Cinn-AgNP-PI.

**Figure 3 pharmaceutics-12-00361-f003:**
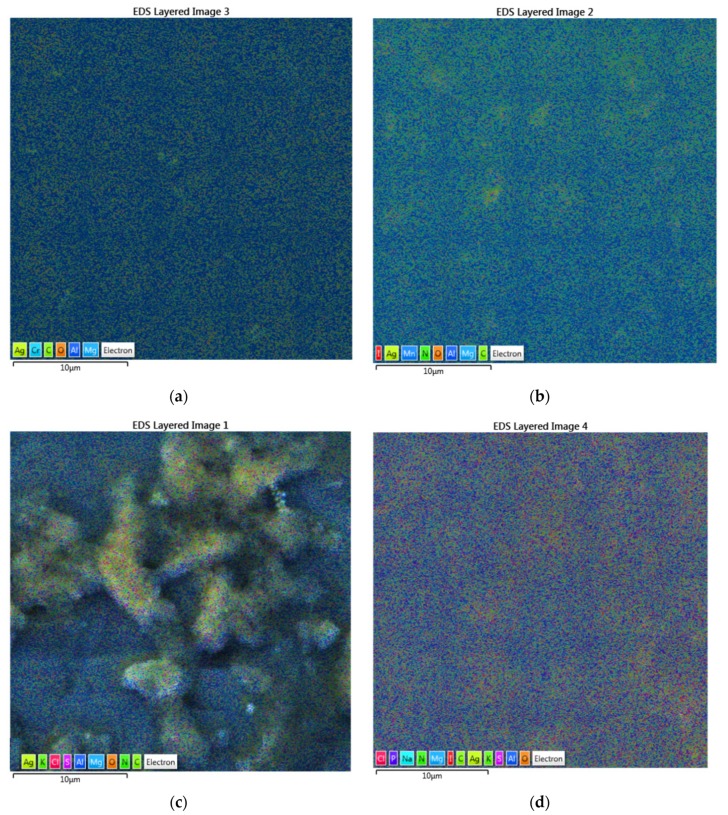
EDS layered images of the four samples. From up to down: (**a**) TCA-AgNP; (**b**) Cinn-AgNP; (**c**) TCA-AgNP-PI; (**d**) Cinn-AgNP-PI.

**Figure 4 pharmaceutics-12-00361-f004:**
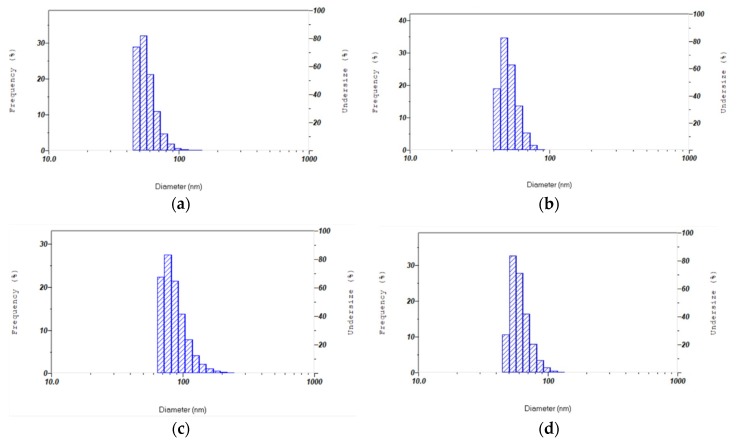
Dynamic light scattering (DLS) analysis of the four samples. From left to right: (**a**) TCA-AgNP; (**b**) TCA-AgNP-PI; (**c**) Cinn-AgNP; (**d**) Cinn-AgNP-PI.

**Figure 5 pharmaceutics-12-00361-f005:**
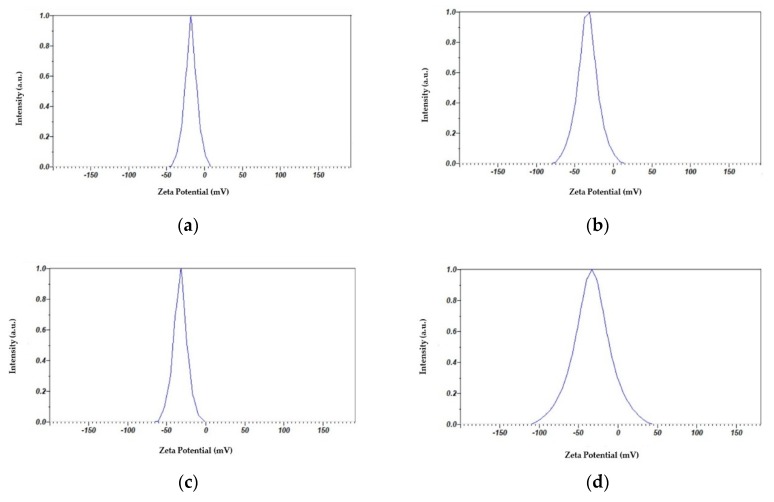
Zeta-Potential analysis of the four samples. From left to right: (**a**) TCA-AgNP; (**b**) TCA-AgNP-PI; (**c**) Cinn-AgNP; (**d**) Cinn-AgNP-PI.

**Figure 6 pharmaceutics-12-00361-f006:**
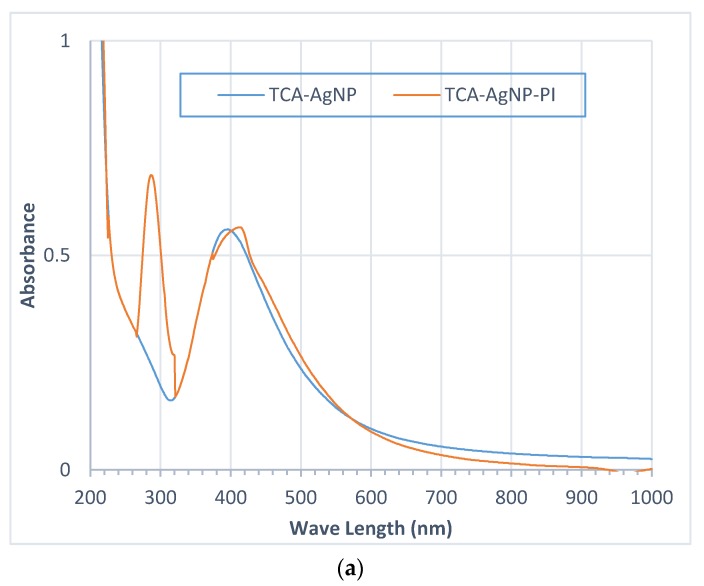
Ultraviolet-visible (UV-vis) spectrometric analysis of the four samples. From up to down: (**a**) TCA-AgNP and TCA-AgNP-PI; (**b**) Cinn-AgNP and Cinn-AgNP-PI.

**Figure 7 pharmaceutics-12-00361-f007:**
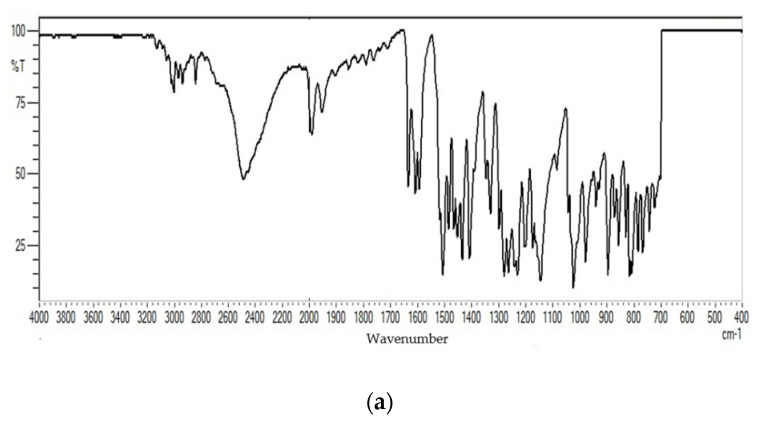
Fourier-transform-infrared (FT-IR) spectrometric analysis of the four samples. From up to down: (**a**) TCA-AgNP; (**b**) TCA-AgNP-PI; (**c**) Cinn-AgNP; (**d**) Cinn-AgNP-PI.

**Figure 8 pharmaceutics-12-00361-f008:**
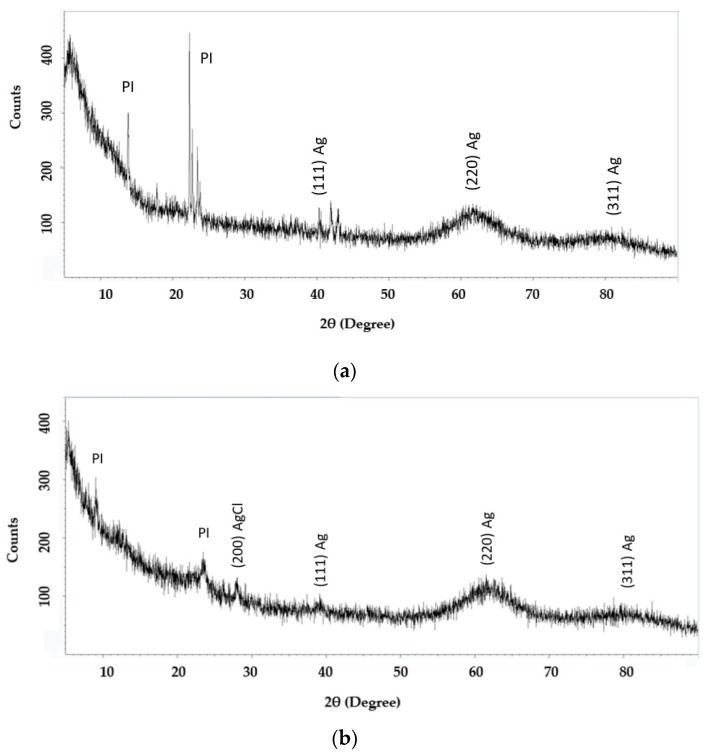
X-ray diffraction (XRD) analysis of the two samples. From up to down: (**a**) TCA-AgNP-PI; (**b**) Cinn-AgNP-PI.

**Figure 9 pharmaceutics-12-00361-f009:**
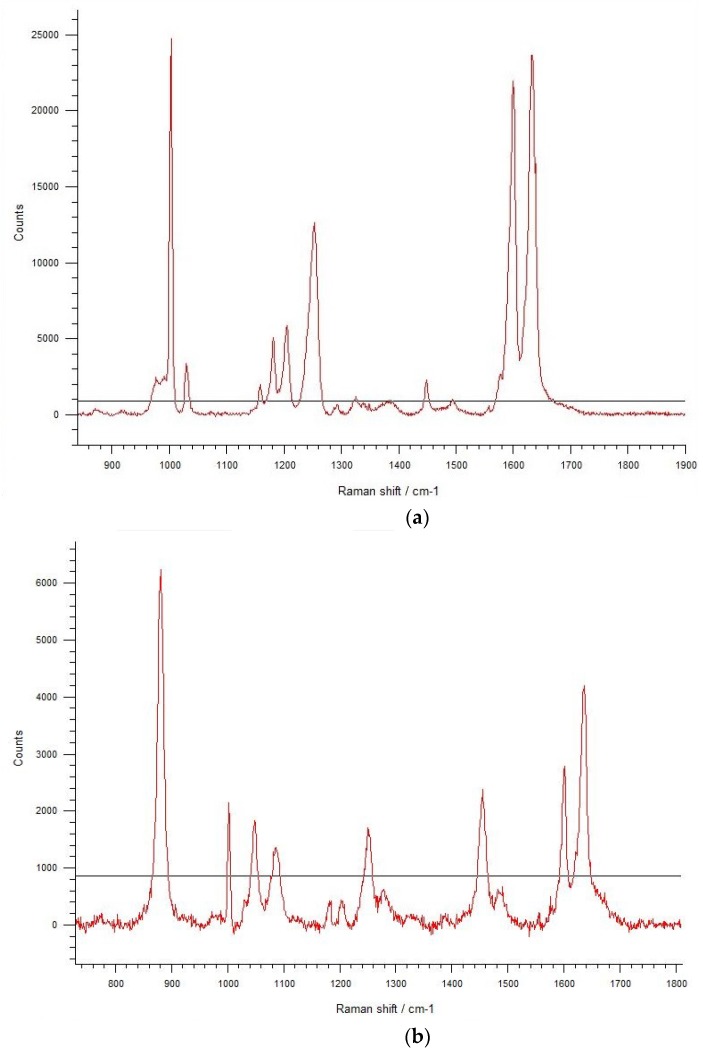
Surface-Enhanced Raman Spectroscopy (SERS) analysis of the four samples. From left to right: (**a**) TCA-AgNP; (**b**) TCA-AgNP-PI; (**c**) Cinn-AgNP; (**d**) Cinn-AgNP-PI.

**Figure 10 pharmaceutics-12-00361-f010:**
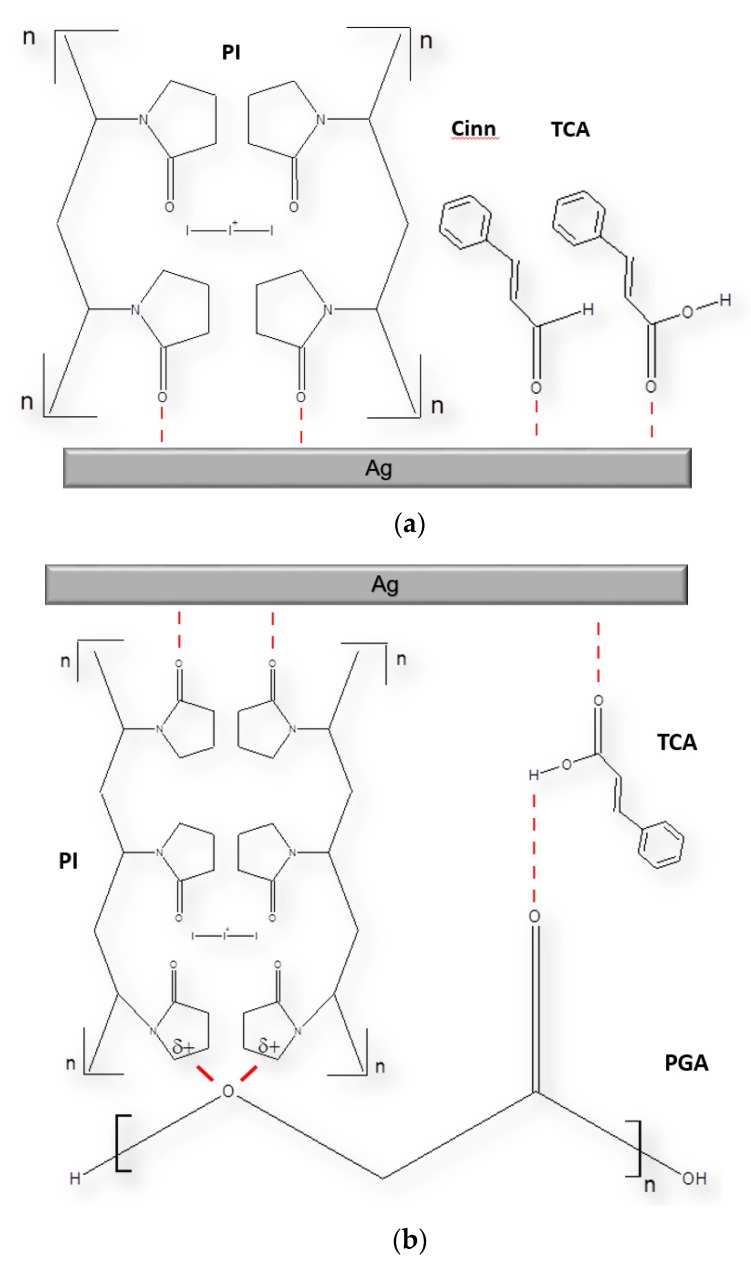
Scheme of physical adsorption in four samples on AgNP. From left to right: (**a**) TCA, Cinn and PI; (**b**) TCA, PI on PGA suture.

**Figure 11 pharmaceutics-12-00361-f011:**
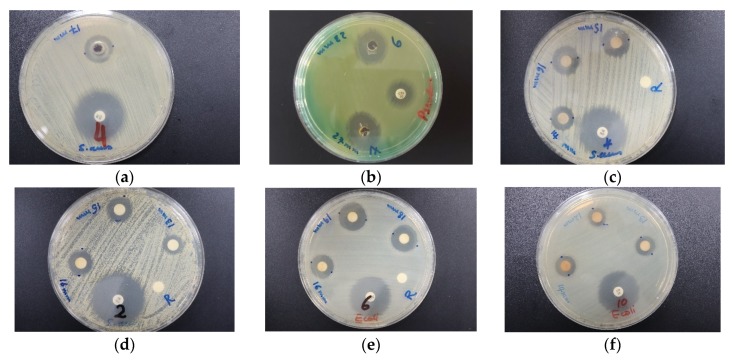
Antimicrobial agar plate methods on AgNP compounds with positive controls (antibiotic). From left to right: (**a**) Agar well diffusion of TCA-AgNP against *S. aureus ATCC 25923*; (**b**) TCA-AgNP-PI against *P. aeruginosa WDCM 00026*. From (**c**–**f**) disc diffusion methods: (**c**) TCA-AgNP against *S. aureus ATCC 25932*; (**d**) TCA-AgNP against *S. aureus ATCC 25932*; (**e**) TCA-AgNP-PI against *E. coli WDCM 00013*; (**f**) Cinn-AgNP-PI against *E. coli WDCM 00013*.

**Figure 12 pharmaceutics-12-00361-f012:**
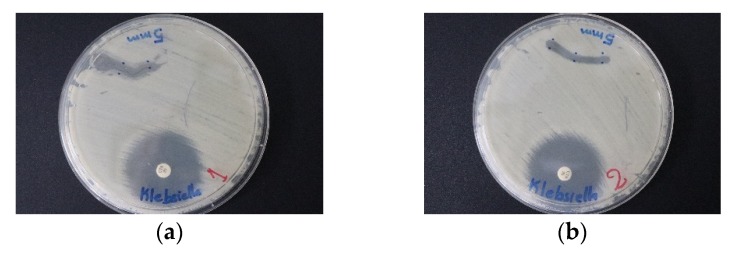
Antimicrobial agar plate methods on dip coated PGA sutures with positive control antibiotic cefotaxime against *K. pneumoniae WDCM 00097*. From left to right: (**a**) Cinn-AgNP-PI; (**b**) TCA-AgNP-PI dip-coated PGA; (**c**) TCA-AgNP; (**d**) Cinn-AgNP-PI.

**Figure 13 pharmaceutics-12-00361-f013:**
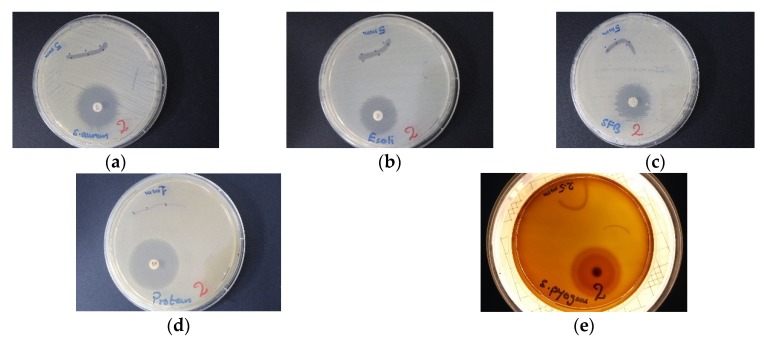
Antimicrobial agar plate methods of TCA-AgNP-PI dip-coated PGA with positive controls (antibiotics). From left to right: against (**a**) *S. aureus ATCC 25923*; (**b**) *E. coli WDCM 00013*; (**c**) clinical sample *Bacillus subtilis*; (**d**) *P. mirabilis ATCC 29906*; (**e**) *S. pyogenes ATCC 19615*.

**Figure 14 pharmaceutics-12-00361-f014:**
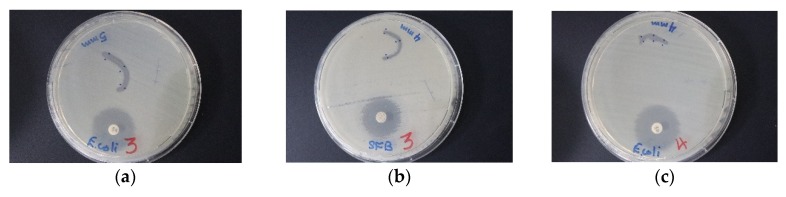
Antimicrobial agar plate methods of dip-coated PGA sutures. From left to right: TCA-AgNP against (**a**) *E. coli WDCM 00013*; (**b**) clinical sample *Bacillus subtilis*; and (**c**) Cinn-AgNP-PI against *E. coli WDCM 00013*.

**Table 1 pharmaceutics-12-00361-t001:** Zeta potential measurements and DLS results for each sample of AgNPs.

Capping Agent Used	Zeta-Potential (mV)	Particle Size Mean (nm)	Poly-Dispersity Index (PDI)
TCA-AgNP	−17.9	61.3 ± 11.2	0.390
TCA-AgNP-PI	−33.3	57.2 ± 10.4	0.464
Cinn-AgNP	−32.9	88.1 ± 20.6	0.376
Cinn-AgNP-PI	−33.1	51.5 ± 7.8	0.486

**Table 2 pharmaceutics-12-00361-t002:** Antimicrobial activity of selected antibiotics (A), TCA-AgNP (1), TCA-AgNP-PI (2), Cinn-AgNP (3), Cinn-AgNP-PI (4), and dilutions. ZOI [mm] against microbial strains by agar-well- (W) and disc dilution assay (D).

Strain	Anti-Biotic	A	1^W^	2^W^	3^W^	4^W^	1^D^	2^D^	3^D^	4^D^	1^D*^	2^D*^	3^D*^	4^D*^
*S. pneumoniae ATCC 49619*	G	21	16	17	0	0	14	13	0	13	10	10	0	0
*S. aureus ATCC 25923*	G	27	17	21	14	20	16	15	10	13	14	10	9	12
*E. faecalis ATCC 29212*	CTX	25	13	15	10	11	12	12	0	12	9	0	0	0
*P. aeruginosa WDCM 00026*	CTX	20	20	27	15	15	20	20	10	14	13	13	0	11
*E. coli WDCM 00013*	G	23	17	17	14	14	17	18	11	12	13	13	10	10
*C. albicans WDCM 00054*	NY	16	18	13	0	15	13	0	0	0	10	0	0	0

^W^ Well diffusion studies with 72 µl of 50 µg/mL compounds 1–4. ^D^ Disc diffusion studies (6 mm disc impregnated with 2 mL of 50 µg/mL and ^D*^ 2 mL of 12.5 µg/mL of compounds 1–4). G Gentamicin (30 µg/disc). CTX (Cefotaxime) (30 µg/disc). NY (Nystatin) (100 IU). Grey shaded area represents Gram-negative bacteria. 0 = Resistant. No statistically significant differences (*p* > 0.05) between row-based values through Pearson correlation.

**Table 3 pharmaceutics-12-00361-t003:** Antimicrobial activity of selected antibiotics (A), TCA-AgNP (1), TCA-AgNP-PI (2), Cinn-AgNP (3), Cinn-AgNP-PI (4) and their dip-coated sutures (S). ZOI [mm] against microbial strains by diffusion assay (S).

Strain	Anti-Biotic	A	1^S^	2^S^	3^S^	4^S^
*S. pneumoniae ATCC 49619*	G	21	0	0	4	0
*S. aureus ATCC 25923*	G	27	1	5	1	1
*E. faecalis ATCC 29212*	CT	25	0	0	0	0
*S. pyogenes ATCC 19615*	C	25	2	2.5	0.5	0
*B. subtilis*	S	20	4	5	0	0
*P. aeruginosa WDCM 00026*	CTX	20	5	5	3	1
*E. coli WDCM 00013*	G	23	5	5	0	4
*K. pneumoniae WDCM 00097*	CTX	35	4	5	5	1
*P. mirabilis ATCC 29906*	G	35	1	1	0	1
*C. albicans WDCM 00054*	NY	16	0	0	0	0

^S^ Diffusion studies (2.5 cm PGA sutures impregnated with 50 µg/mL compounds 1–4). G Gentamicin (30 µg/disc). C Chloramphenicol (10 µg/disc). S Streptomycin (10 µg/disc). CTX Cefotaxime (30 µg/disc). NY Nystatin (100 IU). Grey shaded area represents Gram-negative bacteria. 0 = Resistant. No statistically significant differences (*p* > 0.05) between row-based values through Pearson correlation.

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
