# Peer review of "“Smart” Antimicrobial Nanocomplexes with Potential to Decrease Surgical Site Infections (SSI)"

_pharmaceutics, 2020, doi:10.3390/pharmaceutics12040361_

Round 1

Reviewer 1 Report

This manuscript describes the synthesis and study Ag NPs synthesized using plant extract (cinnamon) or TGA and then combined with povidone-iodine (PI). The authors studied the size, structure and biocidal properties of the synthesized Ag NPs. The paper is well written and the study fits with the scope of Pharmaceuticals. However, most of the figures do not meet the required quality for being published. The graphs are hardly readable (DLS, FT-IR, XRD, Raman) and the images are of poor quality (SEM, EDS). The authors should also reduce the discussion, which is too long and summarize their results more precisely. It is not necessary to repeat the introduction in the discussion part. The manuscript needs to be completely revised before possible acceptance for publication.

  • In the introduction, the authors should mention recent reviews about nanoparticles synthesized using plant extracts (doi.org/10.1016/B978-0-323-51254-1.00014-2). The authors should also mention that usually, plant extract synthesis induces a broad size distribution. Similar articles as the one provided should be sought.
  • SEM images should be improved, the scale bar is not visible and the quality is low.
  • EDS images should be improved, the contrast should be adjusted.
  • Figure caption figure 9, I guess the authors mean Raman and not FT-IR.
  • Raman of figure 9d should be explained.
  • The authors measure the composition of the sample, but they can detect aluminum from the sample holder. What is the accuracy of the measure? The authors should add an error bar.
  • DLS show that the size of Cinn-Ag NPs decreases after PI addition. It seems that PI react with cinnamon extract and removes the coating from the Ag NPs. This should be explained. Similar decrease, but at lower level is visible with TCA-Ag NPs.
  • In UV-Vis spectra, PI peak is not visible in Cinn-AgNP-PI sample. This means that no PI attached on the sample. The authors should comment. This can be discussed with Raman results.
  • The authors should add FT-IR spectra of pure TCA, cinnamon extract and PI in supporting information; otherwise it is difficult to compare and discuss the results.
  • The quality of XRD graph should be improved. It is not possible to read abscissa (2theta). The XRD peaks should be identified and indexed. In fig 8a, the broad peak is certainly (220) of Ag metal structure. However in figure 8b, Ag metal and AgCl structure are both visible. There is many study reporting the presence of AgCl secondary phase in Ag NPs samples synthesized using plant extract. Please look for relevant references.
  • Table 2 and 3 should be improved.

Author Response

Thank you very much for your valuable input !

Best regards

Zehra

Reviewer 2 Report

The present manuscript explains the smart Nanomaterials for Surgical instruments. Manuscript is scientifically sound and I recommend acceptance after minor comments 

  1. Change the title, to more suitable one
  2. Give the graphical abstract which catch the attention of readers
  3. Please provide the TEM and XPS analysis for more understanding of materials.
  4. Explain how material is bound to surgical instruments 

Author Response

Dear Reviewer,

thank you very much for your valuable comments.

Please see the attached document

Best regards

Zehra

Reviewer 3 Report

The manuscript is interesting and well written.  A revision of the graphics is required throughout the whole document.

Figure 1 – The size of each panel (a,b,c, and d ) should be even

Figure 3 – the size of each photo should be increased. It would look better if the Authors rearrange the photos (two photos in the upper and another two photos in the bottom part of the figure)

Figure 6 – the panels a and b  should be merged so that all the UV vis spectra are presented in a single graph. Also, the spectra could be normalized (e.g to the value of 1 at the absorption maxima.)

Figure 7 – IR spectra  - the quality of figure is poor. It would look much better if the figure was presented as stacked spectra. This would allow for the comparison of main bands. Also, the units at the x axis (wavenumbers) should be clearly visible.

Figures 4, 5, 8 and 9 –The fonts of units at the axes should be increased. Currently it is not possible to read the values.

Figures 10, 11, 12, and 13 – the figures sizes should be increased so that the inhibition zones are visible clear and sharp

I recommend the minor revision of the manuscript

Author Response

(The authors gave the same response as above.)

Round 2

Reviewer 1 Report

1) The SEM images are still not readable. I hope this problem is solved during the typesetting.

2) XRD patterns are still not indexed and too noisy. I also see see certain peaks that are artefacts.

Author Response

Dear Reviewer,

Thank you again for your interesting comments.

Best regards

Zehra
